# CRISPR/Cas9-Mediated genomic knock out of tyrosine hydroxylase and yellow genes in cricket *Gryllus bimaculatus*

**Yun Bai**[1], **Yuan He**[1], **Chu-Ze Shen**[2], **Kai Li**[1]*, **Dong-Liang Li**[1]*, **Zhu-Qing He**[1]*

**1** School of Life Science, East China Normal University, Shanghai, China, **2** College of Life Sciences, Beijing Normal University, Beijing, China

* zqhe@bio.ecnu.edu.cn (ZQH); lidongliang@bio.ecnu.edu.cn (DLL); kaili@admin.ecnu.edu.cn (KL)

**Data Availability Statement:** All relevant data are within the paper and its Supporting information files.

## Abstract

*Gryllus bimaculatus* is an emerging model organism in various fields of biology such as behavior, neurology, physiology and genetics. Recently, application of reverse genetics provides an opportunity of understanding the functional genomics and manipulating gene regulation networks with specific physiological response in *G. bimaculatus*. By using CRISPR/Cas9 system in *G. bimaculatus*, we present an efficient knockdown of *Tyrosine hydroxylase* (*TH*) and *yellow-y*, which are involved in insect melanin and catecholamine-biosynthesis pathway. As an enzyme catalyzing the conversion of tyrosine to 3,4-dihydroxyphenylalanine, *TH* confines the first step reaction in the pathway. Yellow protein (dopachrome conversion enzyme, DCE) is also involved in the melanin biosynthetic pathway. The regulation system and molecular mechanism of melanin biogenesis in the pigmentation and their physiological functions in *G. bimaculatus* hasn't been well defined by far for lacking of *in vivo* models. Deletion and insertion of nucleotides in target sites of both *TH* and *Yellow* are detected in both $F_0$ individuals and the inheritable $F_1$ progenies. We confirm that *TH* and *yellow-y* are down-regulated in mutants by quantitative real-time PCR analysis. Compared with the control group, mutations of *TH* and *yellow-y* genes result in defects in pigmentation. Most $F_0$ nymphs with mutations of *TH* gene die by the first instar, and the only adult had significant defects in the wings and legs. However, we could not get any homozygotes of *TH* mutants for all the $F_2$ die by the first instar. Therefore, *TH* gene is very important for the growth and development of *G. bimaculatus*. When the *yellow-y* gene is knocked out, 71.43% of *G. bimaculatus* are light brown, with a slight mosaic on the abdomen. The *yellow-y* gene can be inherited stably through hybridization experiment with no obvious phenotype except lighter cuticular color. The present loss of function study indicates the essential roles of *TH* and *yellow* in pigmentation, and *TH* possesses profound and extensive effects of dopamine synthesis in embryonic development in *G. bimaculatus*.

## Introduction

Crickets are typical singing insects belonging to Orthoptera with over 6000 extant species. They have been used as a classical system on speciation and acoustic communication [1, 2]. To

**Funding:** National Natural Science Foundation of China (No. 31801997) Natural Science Foundation of Shanghai (19ZR1416100) NO - Include this sentence at the end of your statement: The funders had no role in study design, data collection and analysis, decision to publish, or preparation of the manuscript.

**Competing interests:** The authors have declared that no competing interests exist.

study these issues, a representative model species needs to be selected. Ease of breeding [3], large number of eggs [3], detailed embryo development stage tables [4] and assembled and annotated genome [5] are the advantages of *G. bimaculatus* (Orthoptera: Gryllidae). In addition, it has been found that the isolated protein of *G. bimaculatus* can be used as an osteoinductive material for the fabrication of osteoinductive scaffolds or as an alternative to expensive protein supplements to treat bone-related diseases [6]. Thus, *G. bimaculatus* is an emerging model organism in various fields of biology such as behavior, neurology, physiology and genetics [3, 7]. It has been becoming a highly amenable hemimetabolous insect in laboratory [8].

The reverse genetics technology is essential in the development and physiology research of *G. bimaculatus*. At present, there are mainly RNA interference (RNAi), TALENs and the Clustered Regularly Interspaced Palindromic Repeats/CRISPR-associated protein 9 (CRISPR/Cas9) gene editing technology in the research methods of gene function of *G. bimaculatus*. For example, RNAi was used to study tissue-specific regeneration molecular pathways such as JAK/STAT signaling pathways involved in the legs regeneration of *G. bimaculatus* [9]. However, RNAi efficiency can vary between different insect groups, developmental stages or tissues and due to its transient characteristic, it might not be suitable for studying some candidate genes [10]. In the study by Watanabe *et al.*, the application of TALENs in *G. bimaculatus* has been described [11]. Although TALENs are simpler in design and have higher specificity, they have certain cytotoxicity and the module assembly process is cumbersome. In previous studies, the mutation rate of knocking out the *lac2* gene in *G. bimaculatus* was 92% by CRISPR/Cas9 mRNA gene editing system [5]. Stable mutant lines were also generated by injection of Cas9 protein in studies of gene function in *G. bimaculatus* wing development [12].

The cuticle hardening process is very complex and insect cuticle is soft and pale when it is initial synthesized [13]. Tyrosine hydroxylase (TH) is the initial rate-limiting synthetase involved in the biosynthesis of 3,4-dihydroxyphenylalanine (DOPA) during the sclerotization process [14, 15]. DOPA is a quinone melanin precursor essential for exoskeletal pigmentation and cuticle sclerosis [16, 17]. Ddc is a pyridoxal-5-phosphate-dependent enzyme that catalyzes the conversion of DOPA to dopamine. Dopamine is an important neuro-transmitter [18, 19]. Insect major royal jelly proteins (MRJP) or Yellow proteins contain an approximately 300 amino acid-long MRJP domain. They were initially identified in the royal jelly proteins that play a central role in honeybee development [20].

In this article, we target the *TH* gene and *yellow-y* gene in the melanin synthesis pathway. A high proportion of *TH* and *yellow-y* mutants was obtained, including insertion and deletion mutation types. We randomly selected some *yellow-y* individuals and wild type (WT) for mating and generation inheritance. We identified mutations and found they were inherited from parents, indicating that the mutations are stable and inheritable. This greatly promotes the operability of the *G. bimaculatus* as a model system in regulating melanin synthesis, and provides a characteristic animal model for studying the critical roles and protective mechanism of cuticular color.

## Materials and methods

### Insects culturing and egg collection

*G. bimaculatus* was cultured at 25˚C-30˚C, 12 h light:12 h dark photoperiod, and 50% relative humidity (RH). They were feed comprehensive rabbit food, pumpkin and apples. The male and female *G. bimaculatus* were reared to adults in individual. Three days before microinjection, three female adults and five male adults were placed in the same 1000 ml plastic box for mating. Three days after mating, spawning device was prepared. Paper towels moistened with tap water with a height of about 1.5–2.5 cm were placed in a 1000 mL plastic box with a height

of 8.5 cm. The humidity of the paper towel was 60%. Each box contained three mated female adults. After 1 hour, the crickets were moved to a new box to continue laying eggs. According to statistics, a cricket could lay about 240 eggs at a time.

## Identification of *G. bimaculatus TH* and *yellow-y*

The amino acid sequence tyrosine hydroxylase is analyzed in combination with some of the *TH* (BAM15632) genes already reported on NCBI and the transcriptome database of *G. bimaculatus* [21]. The target site of *TH* gene is designed in the seventh exon by sequence alignment (Fig 1). The *yellow-y* gene is searched from the insect-genome database (insect-genome.com), the sequence number is GBI_10058-RA, and its target site is located in the first exon (Fig 1). The TH protein sequence is aligned with the homologous sequences of other twenty-three species and the yellow-y protein sequence is compared with the homologous sequences of other species (S1 and S2 Tables). The phylogenetic tree was constructed using neighbor-joining method and the confidence values of the branch topology were measured from 1000 bootstrapping repeats (Fig 2).

## Construction and synthesis of sgRNA

We used the ZiFit (http://zifit.partners.org/zifit/CSquare9Nuclease.aspx) online tool to design sgRNA. To generate sgRNA templates, one unique oligonucleotide (*TH*-sgF / *yellow-y*-sgF) contained a T7 polymerase binding site and target sequence. The sgRNA-R contains a partial sgRNA sequence that overlaps sequences and was annealed to a common oligonucleotide. gRNA scaffold was cloned into pMD 19-T vector. Double strand DNA for specific gRNA synthesis was PCR amplified. The reaction mix was composed of: 0.2 μL of plasmid, 1 μL of gRNA F primer, 1 μL of gRNA R primer, 25 μL of R040A Prime $^{STAR}$ HS Pre Mix (TAKARA), and 22.8 μL of ddH$_2$O. The reaction conditions were as follows: 94˚C of 5 min, (94˚C of 15 s, 56˚C of 15 s, 72˚C of 15 s) for 35 cycles, and then extended at 72˚C for 10 min. Whole product was run on the 1% agarose gel in TAE buffer. The gel bands were removed and purified using the TIANgel Purification kit (TIAN GEN BIOTECH) according to the supplier's instructions. Transcribe sgRNA using MEGAscript ™ T7 kit according to manufacturer's instructions. The reaction liquid consisted of: 2 μL of ATP solution, 2 μL of CTP solution, 2 μL of GTP solution, 2 μL of UTP solution, 2 μL of 10 X Reaction buffer, 2 μL of Enzyme Mix and 8 μL of temple

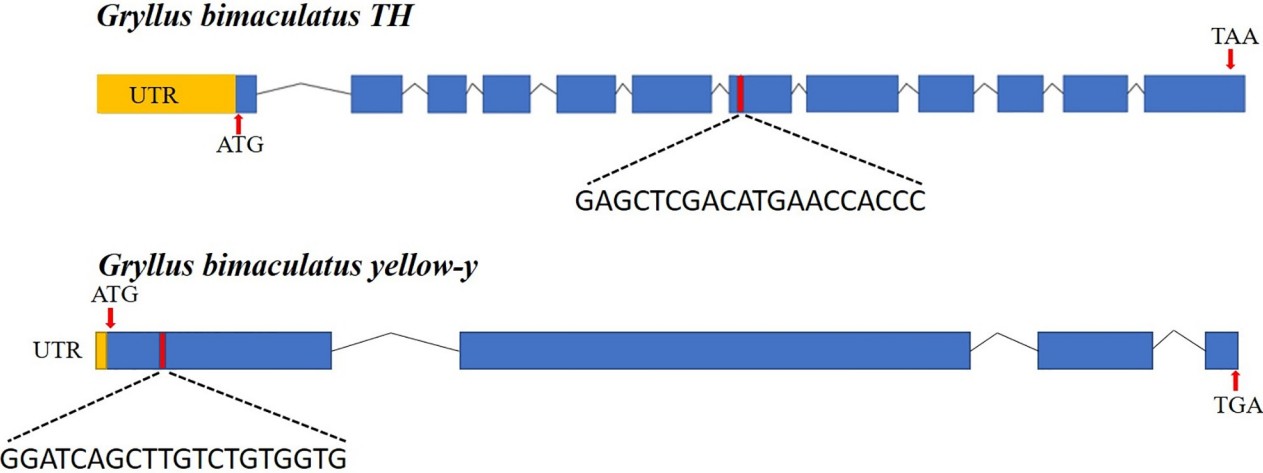

**Fig 1. Mutations resulted from CRISPR/Cas9-mediated disruption of target sites in the *TH* gene and *yellow-y*.** (A) Schematic diagram of the single guide RNA (sgRNA)-targeting sites of *TH*. (B) A schematic of the *yellow-y* gene showing the single guide RNA (sgRNA) target sites. Exons are shown as blue boxes and sgRNA target sites as red.

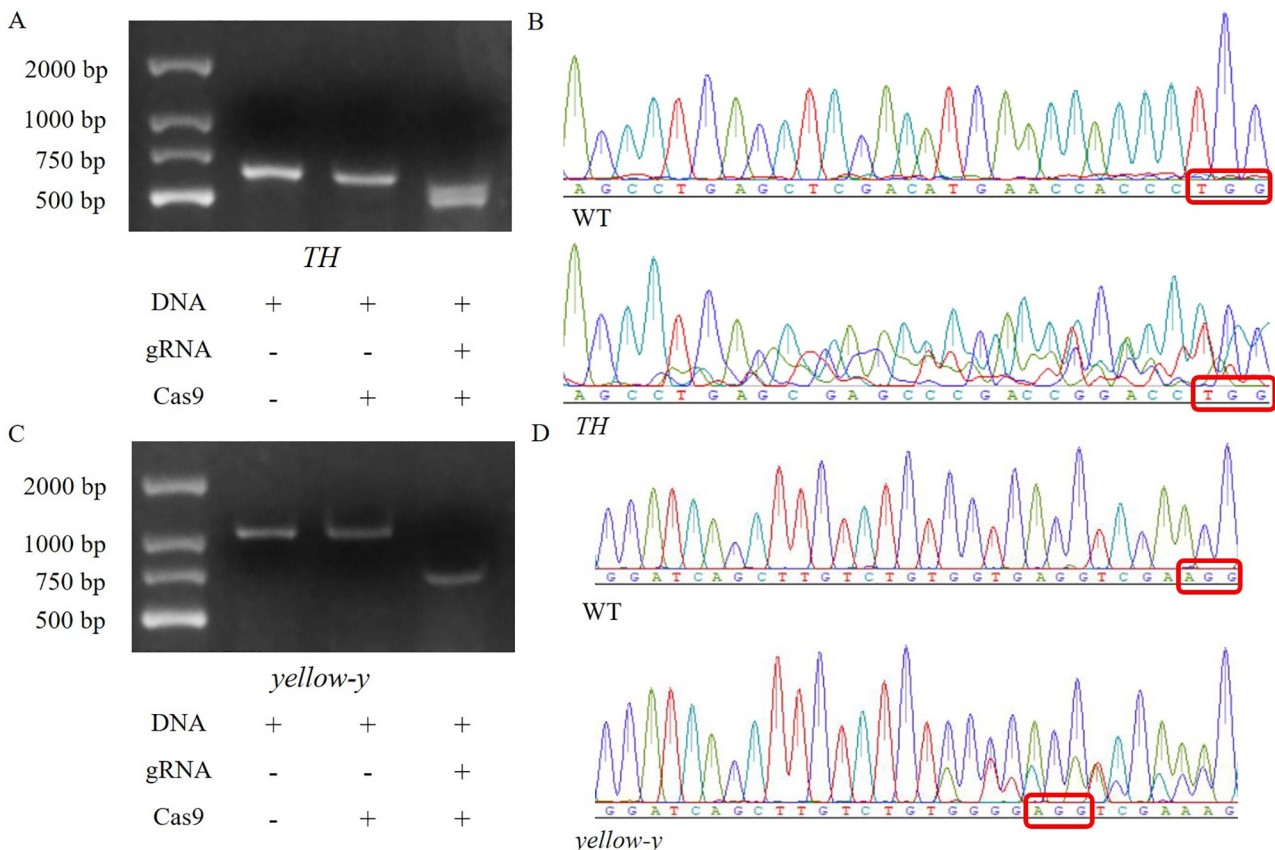

**Fig 2. Verification of the effectiveness of the target.** (A) *In vitro* verification of *TH* sgRNA. (B) DNA sequencing results of WT eggs and eggs injected with *TH* sgRNA and Cas9 protein. (C) *In vitro* verification of *yellow-y* sgRNA. (D) DNA sequencing results of WT eggs and eggs injected with *yellow-y* sgRNA and Cas9 protein. The red box refers to the PAM sequence.

DNA. *In vitro* transcription was carried out for 2 h at 37˚C. 1 μL Turbo DNase was added to degrade the DNA template and incubated at 37˚C for 15 min. sgRNAs were then purified by ethanol precipitation with 3M sodium acetate. The quality of sgRNA was checked with agarose gel electrophoresis, and the concentration was measured with the ultra-micro spectrophotometer (NanoDrop One). sgRNAs were diluted to the concentration of 500 ng/μl. They were aliquoted and stored at −80˚C until further use. All primers are listed in Table 1.

### *In Vitro* verification of sgRNA efficiency

Cas9 protein (E365-01A) was purchased from Novoprotein Company. The final injection concentration was 300 ng/μL. The *in vitro* verification system was composed of: 14 μL of Target gene PCR product, 3 μL of sgRNA, 1 μL of NLS-Cas9 Nuclease and 2 μL of 10 X Reaction Buffer. In the control group, sgRNA was replaced with sterilized water. The reaction conditions were as follows: 37˚C for 30 min, 70˚C for 10 min. Then the cleavage efficiency of sgRNA was detected by 2% agarose gel electrophoresis.

### Microinjection

The eggs of 1 h post spawning were neatly placed in the agar plate model. The needle made by the Micropipette Puller (P-97 Sutter, USA) was cut out of a bevel, sharp like a syringe. The

**Table 1. Primer sequences used in this study.**

| Target | Direction | Sequence 5′ to 3′ |
|---|---|---|
| *TH* (sgRNA) | Forward | TAATACGACTCACTATAGGGTGGTTCATGTCGAGCTCGTTTTAGAGCTAGAAATAGC |
| *yellow-y* (sgRNA) | Forward | TAATACGACTCACTATAGGATCAGCTTGTCTGTGGTGGTTTTAGAGCTAGAAATAGC |
| gRNA | Reverse | AAAAAAAGCACCGACTCGGTGCCAC |
| *TH* (DNA) | Forward | TCGTGGTGTTTCAGACCCCT |
| *TH* (DNA) | Reverse | GTCGAGACCTTGGCAGCTTT |
| *yellow-y* (DNA) | Forward | TCTTCGAGTTCCAACAGCCG |
| *yellow-y* (DNA) | Reverse | AGCTGACTCCCTACCCAGAC |
| *TH* (qPCR) | Forward | TCGGAACTCGACAACTGCAA |
| *TH* (qPCR) | Reverse | TTCGCGGTAGACCTTGTCTG |
| *yellow-y* (qPCR) | Forward | CGTCGGGCTTGTACTGGTAA |
| *yellow-y* (qPCR) | Reverse | GGCAACACCACGGAGAATGT |
| *β-actin* (qPCR) | Forward | TTGACAATGGATCCGGAATGT |
| *β-actin* (qPCR) | Reverse | AAAACTGCCCTGGGTGCAT |
| *Ddc* (qPCR) | Forward | AGCTGGGTGTCGTATGCAAT |
| *Ddc* (qPCR) | Reverse | GCATGTTCAATTCCGGCCAT |

sgRNA and Cas9 protein were mixed according to the system: 3.6 μL of sgRNA, 1.8 μL of Cas9 protein, and 0.6 μL of 10X buffer, then added to the needle, and microinjected with a microinjector (PLI 100 A, Harvard, USA). A mixture of Cas9 protein (300 ng/μL) and *TH* sgRNA (80 ng/μL, 150 ng/μL, 300 ng/μL, 500 ng/μL) was injected into the eggs. The concentration of *yellow-y* sgRNA was: 300 ng/μL and 500 ng/μL. Injected eggs were incubated at 28˚C, 60% humidity and 12:12 h photoperiod and sprayed with sterile water containing 50 U/ml penicillin and 50 μg/ml streptomycin (15070–063, Thermo Fisher) every day.

## Observing and photograph

As previously described, the body color of hatchlings become black after 1 day [4]. The pigmentation process of representative individuals in *TH*, *yellow-y* and WT were photographed just after hatching, at 0.5 h, 1 h, 1.5 h, 3.5 h, 24 h and 5 days after hatching, respectively. Five days after hatch, *TH* mutant, *yellow-y* mutant and WT were photographed by microscope (Leica M125 stereo microscope) after crickets placed on ice for 5 minutes. In addition, pigmented and representative individuals were photographed their heads for comparison. Mutant and WT adults were photographed by camera (Canon PowerShot G7 X Mark III).

## Genomic DNA extraction and identification of mutagenesis

Genomic DNA was extracted from hatched larvae or antennae of adult of the injected eggs with a DNA extraction buffer, incubated with proteinase K, and purified with the AxyPrep ™ Multisource Genomic DNA Miniprep Kit. The PCR conditions were 98˚C for 2 min, followed by 35 cycles of 94˚C for 10 s, 56˚C for 30 s, and 72˚C for 1 min, followed by a final extension period of 72˚C for 10 min. The PCR products were directly sequenced. All primers are listed in Table 1.

Thirteen $F_0$ nymphs injected with *TH* gene, 6 $F_0$ individuals injected with *yellow-y* gene and four $F_1$ individuals of *yellow-y* were randomly selected. These individuals showed double peaks in DNA sequencing around the target sites. PCR products of these individuals were TA cloned to identify the specific type of mutation.

## Quantitative real-time PCR (qPCR) analysis

qPCR was performed to analyze the expression levels of *TH* and *yellow-y* genes with three 1st-instar nymphs of mutant and WT. Taq Pro Universal SYBR qPCR Master Mix (Vazyme) was used for the qPCR assays. The cycling procedure was as follows: initial incubation at 95˚C for 30 s, 40 cycles of 95˚C for 10 s, 60˚C for 30 s, and 95˚C for 15 s, 60˚C for 60 s, 95˚C for 15 s. The actin gene of *G. bimaculatus* was used as reference. All experiments were performed in three biological replicates. All primers are listed in Table 1.

## Germline inheritance

Female and male $F_0$ individuals with either *TH* or *yellow-y* mutations were selected and mate with the WT siblings. Three days after mating, females were placed in the spawning device to lay eggs. Eight days after spawning, each group has 100 of the $F_1$ eggs were randomly selected for DNA sequencing. The rest of the eggs were cultured to adults for strain construction.

## *In vivo* melanization assays

Three 1st-instar nymphs of the same size and significantly lighter cuticles in the *TH* group mutants were ground at 4˚C with 180 μL PBS (pH = 7.0) and centrifuged (500× g for 5 min at −4˚C). The nymphs of WT group were used as the control. A total of 80 μL supernatant (the Bradford method quantitative protein concentration was 6 mg /mL) was transferred into PCR tube. The A490 value was determined to assess the melanin content.

## Dopamine contents and genes expression in cuticle tanning pathway

The dopamine content of *TH* and WT nymphs at the 1st-instar was determined using a commercial kit (Nanjing Jiancheng Institute of Bioengineering, China). Absorbance of 490 nm was detected in Microplate Spectrophotometer. The samples were homogenized in 1 X PBS and centrifuged at 4˚C and 12000 g for 10 min. Supernatants were used to measure the dopamine concentration according to the manufacturer's protocols. The expression level of *Ddc* gene involved in the cuticle tanning pathway was detected. The qPCR primer sets are presented in Table 1. Data were normalized for β-actin, and two independent samples were analyzed using Student's t test.

# Results

## Identification and analysis of *TH* gene and *yellow-y* gene

Genomic structure analysis reveals that the *TH* gene contains twelve exons and eleven introns (Fig 1). The *yellow-y* gene contains four exons and three introns (Fig 1). Twenty-three orthologous protein sequences reported in other insects and *TH* of *G. bimaculatus* are used to construct a phylogenetic tree (S1 Fig). Phylogenetic analysis of the *yellow* gene of *G. bimaculatus* shows that it is closely related to the *yellow-y* gene of other species (S1 Fig).

## The effectiveness of *TH* gene and *yellow-y* gene targeting

The *in vitro* verification (Fig 2; S2 and S4 Figs) shows that *TH* sgRNA and *yellow-y* sgRNA are effective. The target strip can be cut into two segments of different sizes. The original length of *TH* gene fragment is 545 bp, which is digested into 464 bp. The original length of *yellow-y* gene fragment is 1058 bp and is digested into 826 bp. Using Cas9 protein without sgRNA as negative control, non-specific cleavage is excluded.

The results of *TH* gene microinjection (S3 Table) show that the average hatching rate of 80 ng/μL was 13.00%, and the average mutation rate was 69.23%; the average hatching rate of 150 ng/μL was 10.00%, and the average mutation rate was 90.00%; the average hatching rate of 300 ng/μL was 8.00%, and the average mutation rate was 87.50%; the average hatching rate of 500 ng/μL was 11.00%, and the average mutation rate was 90.91%. The DNA sequencing results also indicate that comparing with WT, the mutants show double peaks or random peaks at the target site, indicating the occurrence of mutations (Fig 2). These experimental results fully prove the effectiveness of the target.

In order to determine the specific targeted mutagenesis of the *TH* gene by the CRISPR/Cas9 gene editing system, 13 individuals were randomly selected for TA cloning with DNA sequencing results showing double peaks at target site. The sequencing results show that there are two types of mutations, either deletion or insertion. The specific mutation types are deletion of 6 bp, 21 bp, 4 bp, 5 bp, 1 bp, 3 bp, 5 bp, 4 bp, 2 bp and insertion of 8 bp, 5 bp, 2 bp, 9 bp (Fig 3).

The results of TA cloning of *yellow-y* $F_0$ generation show that the types of mutations including deletion of 3 bp and 4 bp (Fig 3). A female and a male with mutations were randomly selected to cross WT and their offspring were identified. Among the offspring of *yellow-y* ♀ / WT ♂, 90 eggs are sequenced, of which 50 are mutants and 40 are WT. Among the offspring of WT ♀ / *yellow-y* ♂, 94 eggs are sequenced, of which 50 are mutant and 44 are WT (Table 2). These results indicate that the CRISPR/Cas9-mediated genome editing system has reliable functions in crickets and can be genetically inherited.

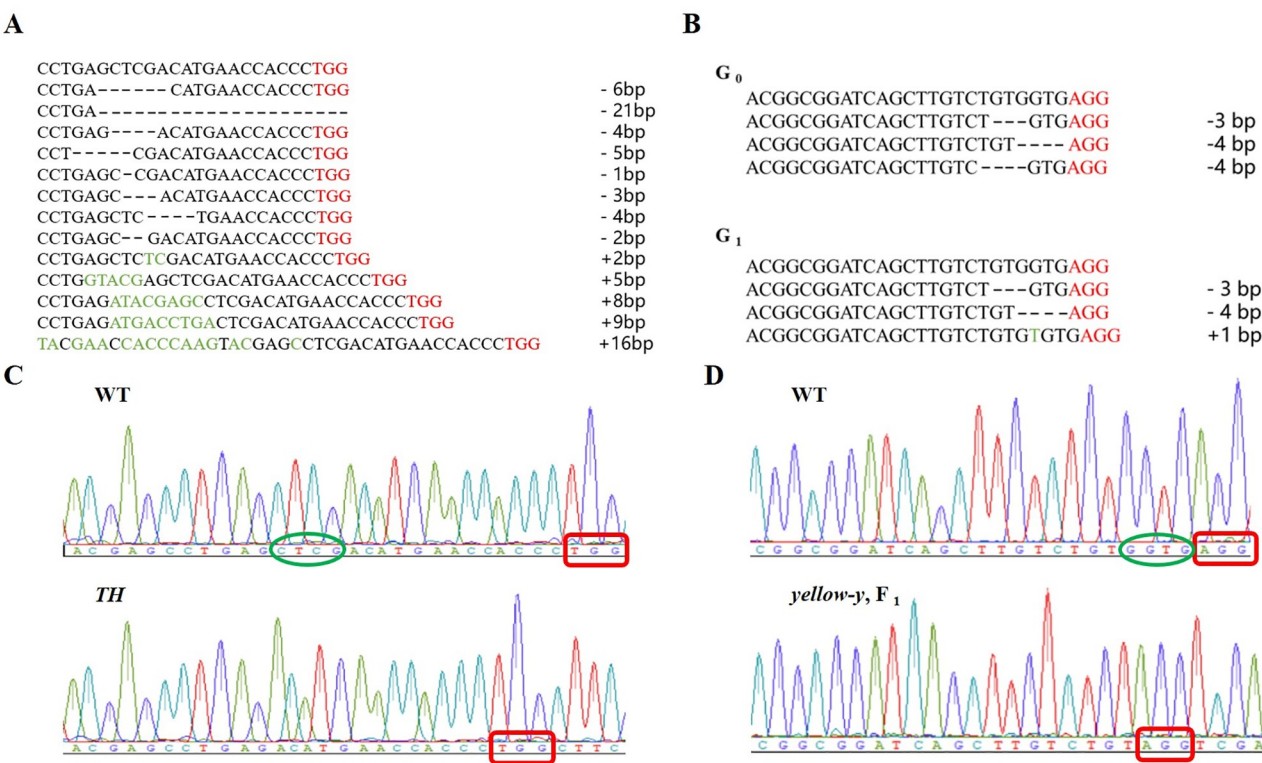

**Fig 3. Mutations resulted from CRISPR/Cas9-mediated disruption of target sites in the *TH* gene and *yellow-y* gene.** (A) Thirteen types of mutations in $F_0$ hatched nymphs of *TH* gene. (B) Three types of mutations in $F_0$ hatched larvae and $F_1$ eggs of *yellow-y*. (C) Below the map is the result of sequencing. (D) DNA sequencing results of WT eggs and eggs injected with *yellow-y* sgRNA and Cas9 protein. There are some deletions and insertions at a single targeting site. The PAM sequence is in red. The green circle shows deleting bases.

**Table 2. The statistical analysis of efficiency for the *yellow-y* heredity of offspring.**

|  | *yellow-y*♂ × WT♀ | WT♂ × *yellow-y*♀ |
|---|---|---|
| Mutant | 50 (53.19%) | 50 (55.60%) |
| WT | 44 (46.81%) | 40 (44.40%) |

We compare the expression levels of *TH* and *yellow-y* in WT and $F_0$ crispants. *TH* and *yellow-y* transcripts are significantly down-regulated in mutants (Fig 4).

## Phenotypic analysis

*TH* gene and *yellow-y* gene are involved in the synthesis pathway of insect melanin [22, 23]. Knocking out the *TH* gene in embryos, the body color of the *G. bimaculatus* nymphs (5 days after incubation) is mosaic-like in varying degrees (Fig 5; S3 Fig). The individuals of missing three base pairs or multiples that show no obvious changes in body color (35.85%), while the individuals with deletion or insertion neither three base pairs or multiples show more obvious changes in body color (64.15%). Most of these hatched nymphs died before or during the first molting. There is only one individual survives to adult. The legs and wings of the adult are severely defective, which interfere with mating (Fig 6). Therefore, there is no stable genetic line yet. We tried to cross with WT to obtain $F_1$ heterozygotes, and the incubation rate of $F_1$ was significantly reduced. The heterozygotes of $F_1$ generation successfully incubated were normal in phenotype. But all the $F_2$ homozygotes died from molting. Therefore, there is no stable genetic line yet.

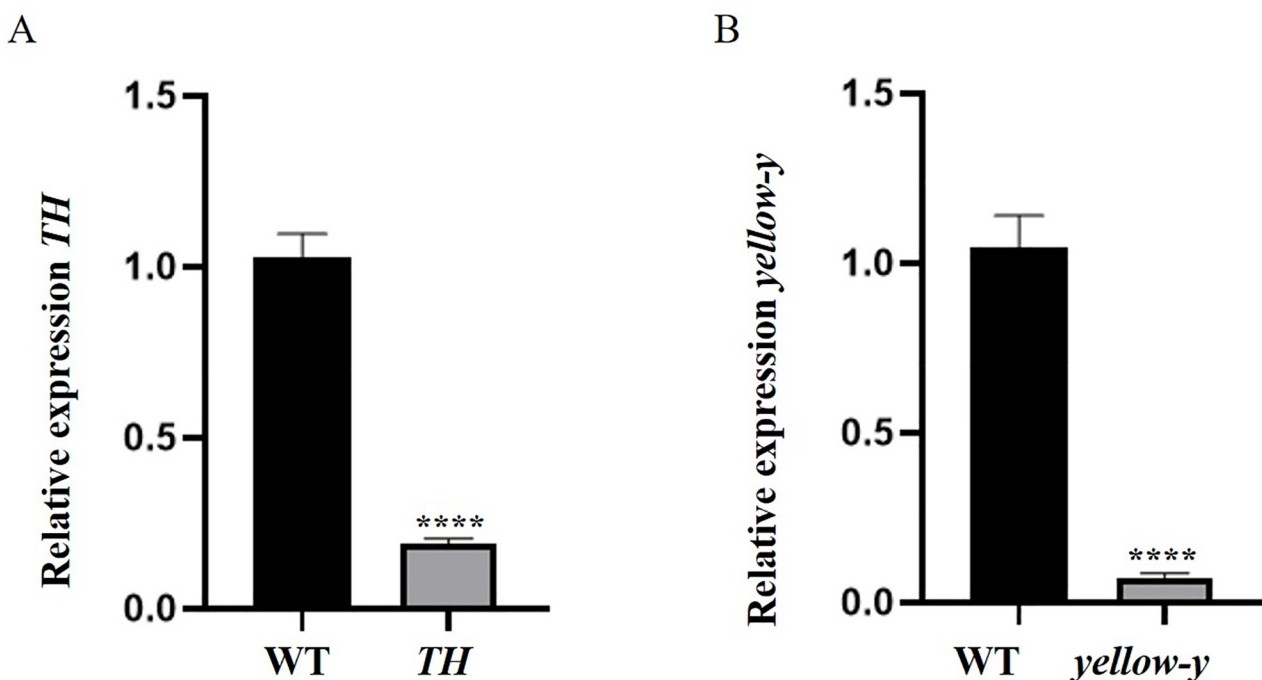

**Fig 4. The relative mRNA expression in WT and mutants.** (A) Decreased mRNA level of *TH* was detected in *TH* mutants. P < 0.0001. ($t$ = 11.9, $df$ = 22) (B) Decreased mRNA level of *yellow-y* was detected in *yellow-y* mutants. P < 0.0001. ($t$ = 10.40, $df$ = 22) The mRNA levels were normalized to beta-actin. The mean of the relative expression of the WT is set to 1. The error bars represent the mean ± SE. Asterisks represent statistically significant differences.

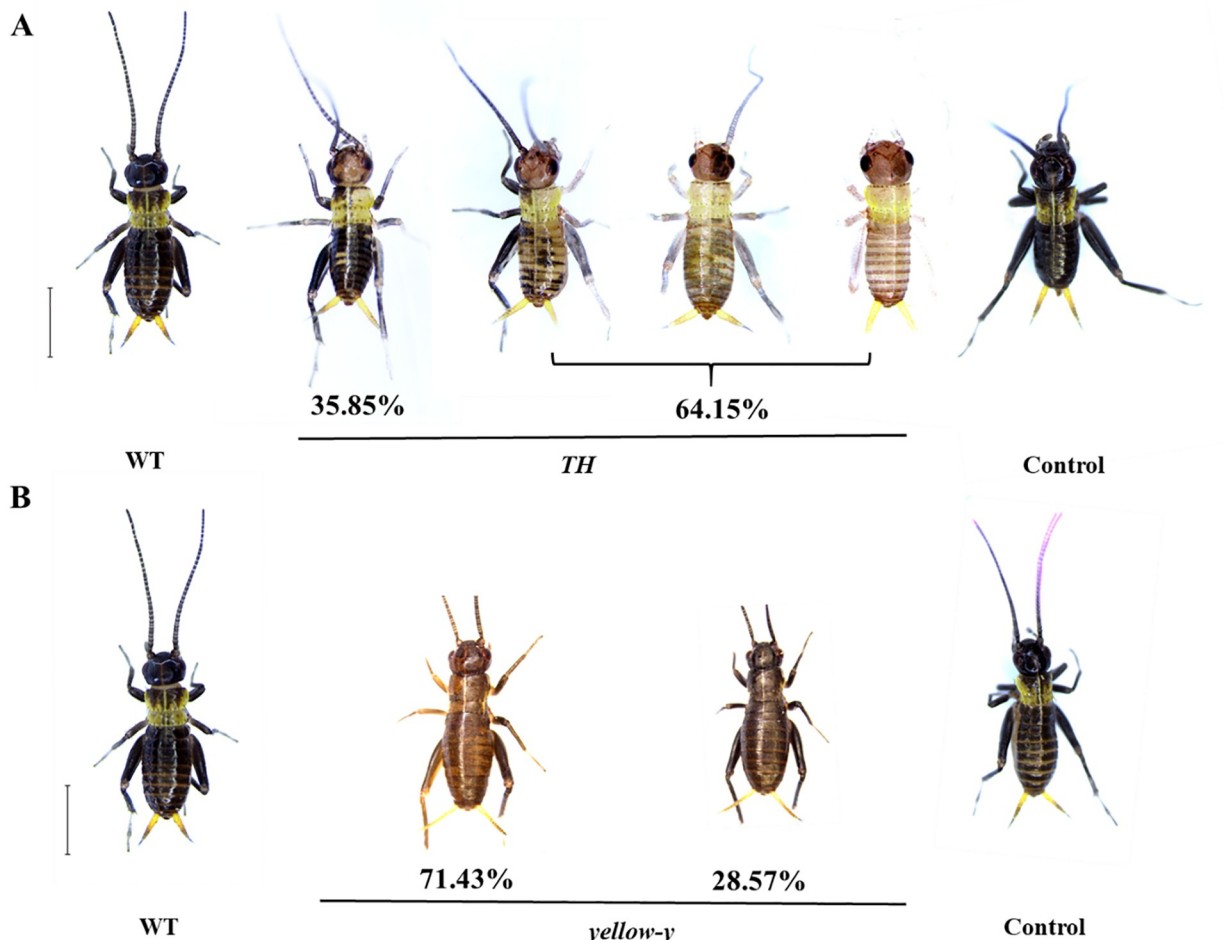

**Fig 5. Phenotypic analysis.** (A) Phenotypes of dysfunction *TH* in *G.bimaculatus*. The WT was observed on the 5th-day of 1st larval instar. (B) Phenotypes of dysfunction *yellow-y* in *G. bimaculatus*. The WT was observed on the 5th-day of 1st larval instar. The mutants are brown mosaic. Control represents an individual injected with Cas9 protein only. Bar = 3mm.

When the *yellow-y* gene is knocked out, 71.43% of *G. bimaculatus* are light brown, with a slight mosaic on the abdomen (S3 Table, Fig 5). This phenomenon could be inherited between germ lines. In the offspring of individuals, the body color of *G. bimaculatus* has two types: brown or black (Fig 7). Mutants accounted for 53.19% in *yellow-y*♂ × WT♀ group and 55.60% in WT♂ × *yellow-y*♀ group (Table 2).

The pigmentation is partially completed after 24 hours after abolishing of the *TH* and *yellow-y* genes, while the pigmentation of WT is completed in 1.5 hours (Fig 8). The heads of *TH* and *yellow-y* genes mutants are mosaic-shaped, with the *TH* mutant having a light brown and black mosaic heads, and the *yellow-y* mutant having a dark brown and black mosaic head (Fig 9).

## Suppression of *TH* impairs *G. bimaculatus* pigmentation and dopamine synthesis

The *in vivo* melanization experiments show that the melanin productions of the 1st-instar nymphs of *TH* mutants are significantly less than that of controls. The values of the absorbance at 490 nm in the control group are: 0.28, 0.192, and 0.24, and those in the experimental group

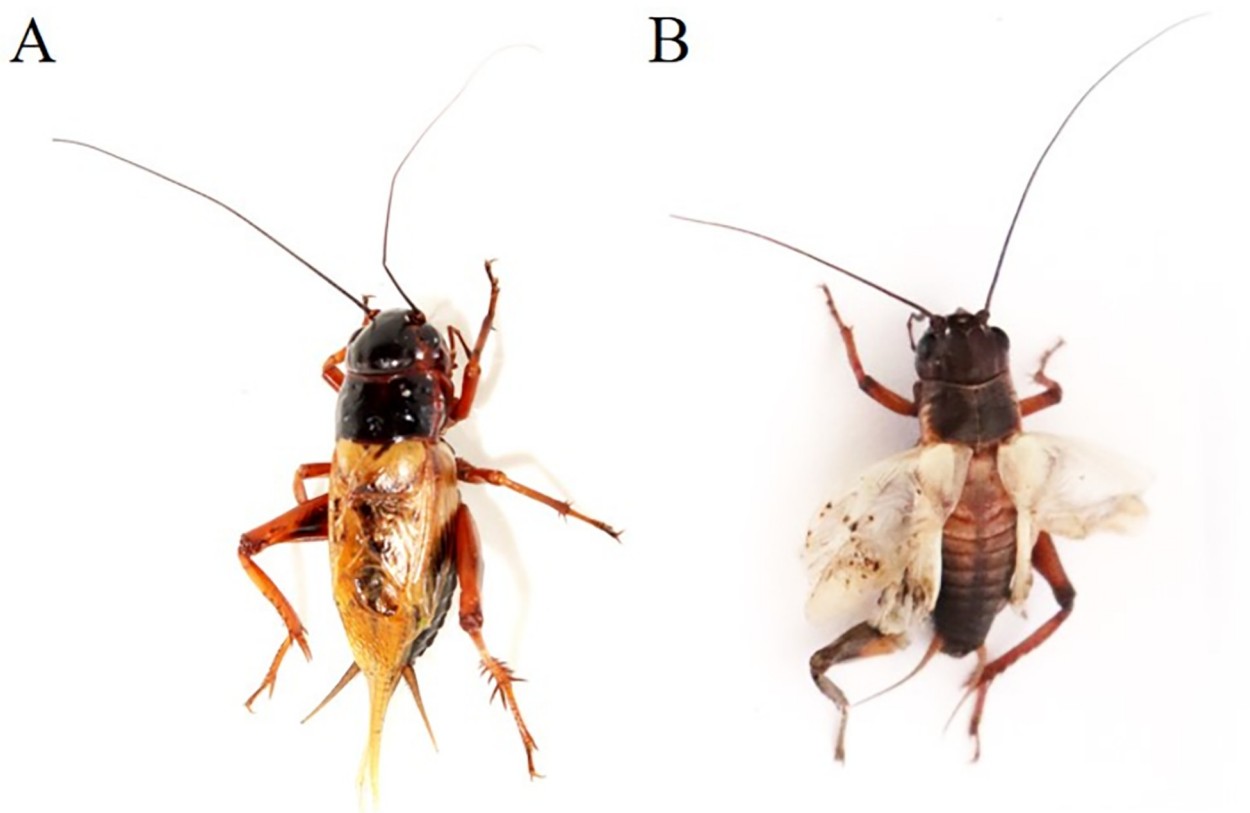

Fig 6. **Phenotypic analysis of adults.** (A) Phenotype of WT adult. (B) Phenotype of *TH* mutant adult.

are: 0.125, 0.125, and 0.142 (Fig 10). In agreement with the blockage of cuticle tanning, the dopamine concentration is significantly decreased (Fig 11A) and the expression level of the *Ddc* gene is also significantly down-regulated in individuals with *TH* knockout (Fig 11B).

## Discussion

In the experiments of the CRISPR/Cas9 gene editing system in *Pyrrhocoris apterus*, Kotwica-Rolinska *et al.* pointed out that Cas9 protein as the preferred choice of this approach [24]. The combination of high concentrations of Cas9 and sgRNA has been reported to result in a higher mutation rate but a lower survival rate [25, 26]. And injection of the sgRNA and Cas9 protein complex into the zygote forming region enabled Cas9 nuclease to edit the gene immediately after injection [27]. Both of Cas9 plasmid and mRNA could not function immediately [28], they need essential time to experience a protein synthesis process in the embryo [28] and may lower the editing efficiency. In our experiments, the mutation rate is similar to that of previous Cas9 protein systems [12] and mRNA [29]. In addition, the efficiency of mutation is stable. The setting of a series of concentration gradients for the *TH* gene target is beneficial to the preliminary study and evaluation of the concentration-dependent effects of Cas9 protein and sgRNA on the mutagenesis efficiency and survival rate of *G. bimaculatus*. However, the study of off-target effects still needs to be carried out and perfected in follow-up experiments. While there are some CRISPR design tools that can predict off-target effects in some genes [30], we cannot rule out the possibility that there are some off-target effects in unassembled duplicate regions.

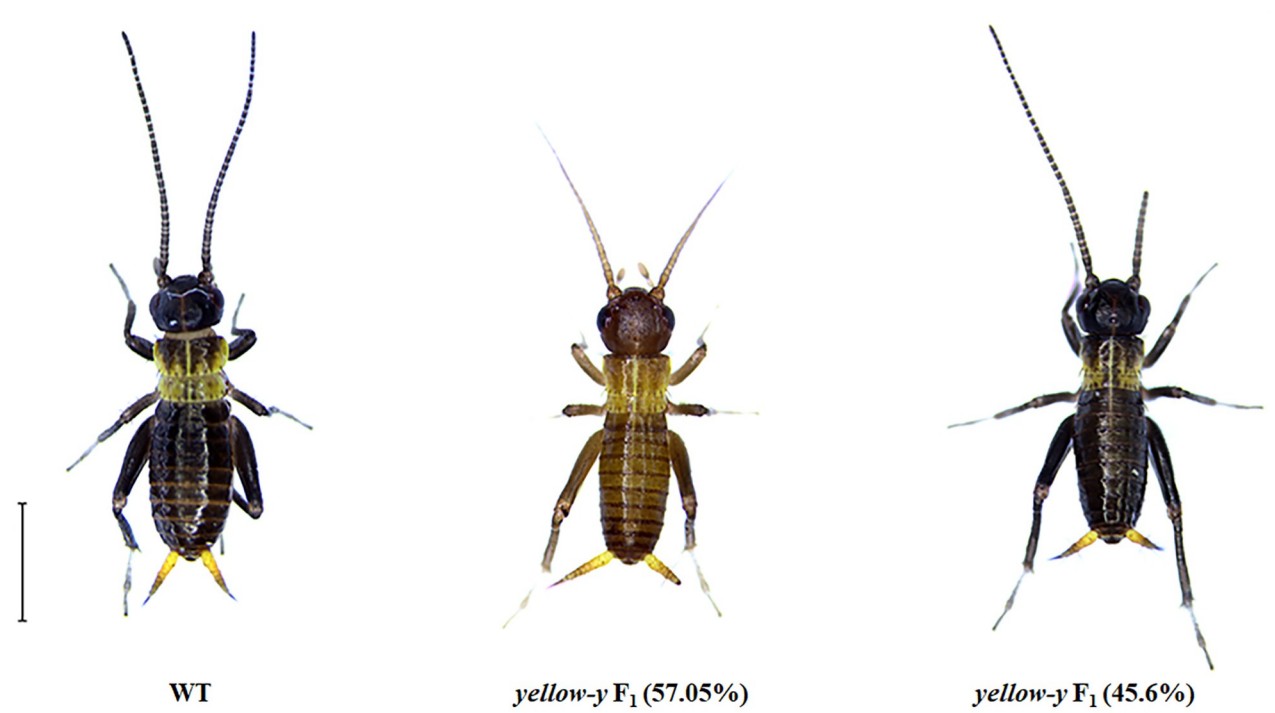

**Fig 7. Phenotype analysis of the F₁ progeny of the *yellow-y* gene.** Brown and black phenotypic differentiation appears in F₁ progeny. The F₁ progeny are viewed on the 8th-day of 1st larval instar. Bar = 3mm.

The cuticle is the outermost layer of an insect's body, and its physical properties, such as elasticity, thickness, strength, and stiffness, are key determinants of maintaining movement, mechanical support, body shape, and normal development [31]. As the insect exoskeleton, the cuticle is also the first protective barrier against various environmental stresses and against the penetration of external compounds such as pathogens, dehydration, physical damage and pesticides [32]. However, the rigid insect exoskeleton prevents further growth of molting and metamorphosis. Therefore, cuticle digestion and biosynthesis in insects must occur periodically. Newly synthesized insect cuticle is known to be pale and soft, and must undergo the tanning process (hardening and pigmentation) within a specific period of time for the insect to growth and development [33]. Melanin and quinone pigments derived from catecholamines (such as dopamine) produced by tyrosine-mediated metabolism of cuticle tanning (pigmentation and sclerotization) play a key role in the darkening and hardening of many insect cuticles [23]. *TH* gene is the initial limiting enzyme for dopamine synthesis. The role of *TH* and *TH* homologues in the melanic pigments of holometabolous and hemimetabolous insects is conservative. In *Agrotis ipsilon*, most CRISPR/Cas9-mediated *AiTh* knockout embryos can develop but cannot hatch. Successfully hatched embryos died within the first day [34]. Injection of *TH* dsRNA or feeding *TH* inhibitors into *Tribolium castaneum*, *Anopheles sinensis* and several fruit flies will severely hinder the larval-pupa deformation and adult emergence, and cause the death of most flies [35–37]. Although the sequence of the *TH* gene has been obtained in the study of the serotonin composition of the *G. bimaculatus* [38], decreased *TH* gene expression levels by RNAi caused wings to become white [39]. Based on gene alignment, we designed a target in the seventh exon of the *TH* gene of *G. bimaculatus* [37]. Our results show that after knocking out the *TH* gene, the body color showed a significant mosaic and most of them died before or during the first molting. Although only one cricket survived as adult, it

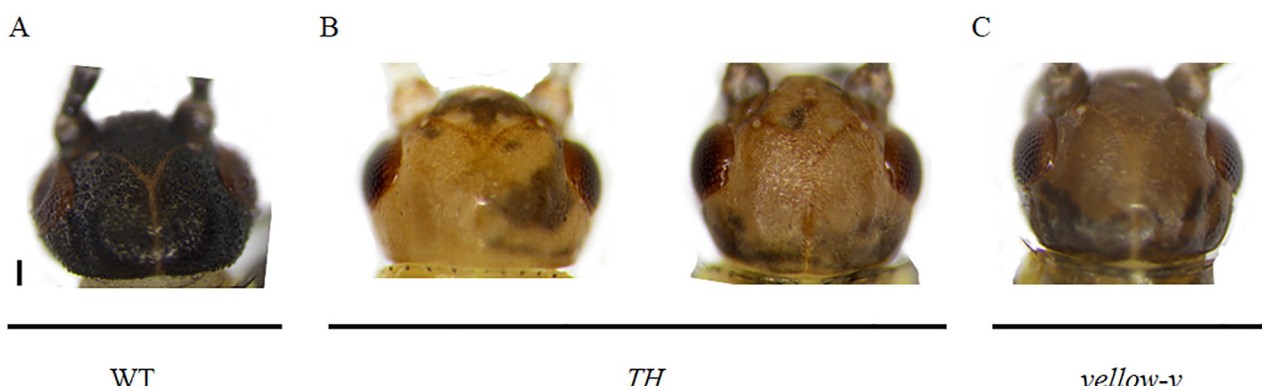

**Fig 8. Pigmentation process in the epidermis of the *G. bimaculatus*.** (A) WT pigmentation process. (B) Pigmentation process in *TH* knockout mutants. (C) Pigmentation process in *yellow-y* mutants. Bar = 1000 um.

**Fig 9. Diagram of pigmentation changes on the head of the *G. bimaculatus*.** (A) WT with black head. (B) *TH* mutant head pigmentation patterns. (C) The head of *yellow-y* mutants are mosaic. Bar = 100 um.

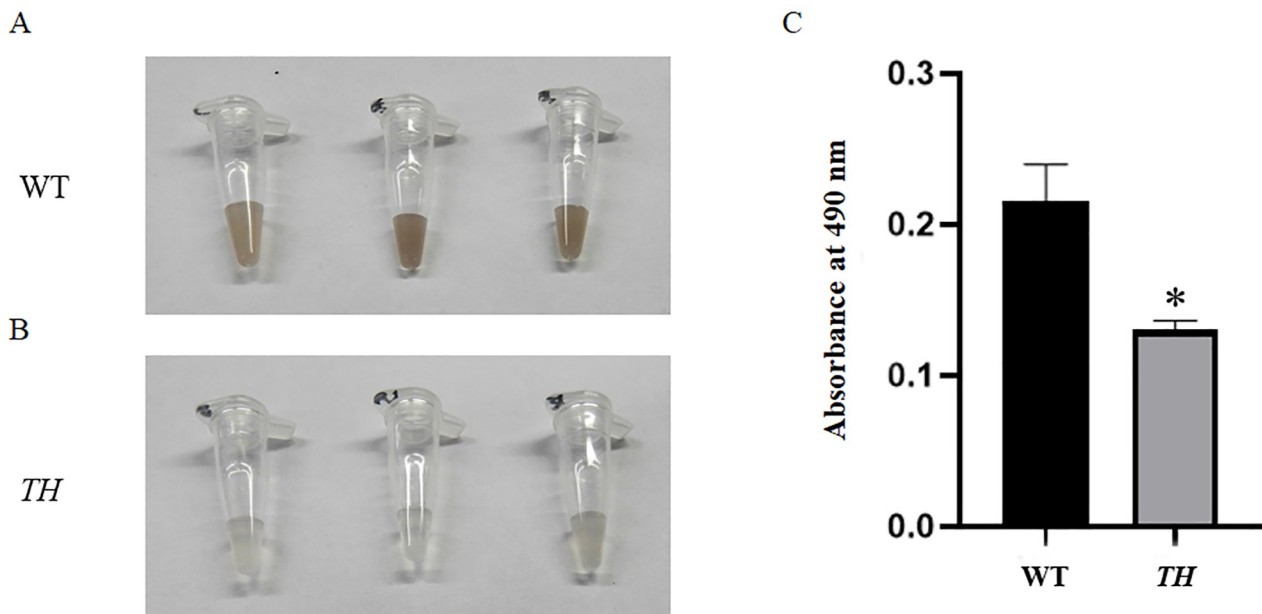

**Fig 10. Effects of *TH* gene knockout on pigmentation.** (A) and (B) present the degree of sample melanism of the *in vivo* melanization. (C) Absorbance at 490 nm for each sample. Three replications are conducted, and the data are presented as mean ± SE. P = 0.0149.

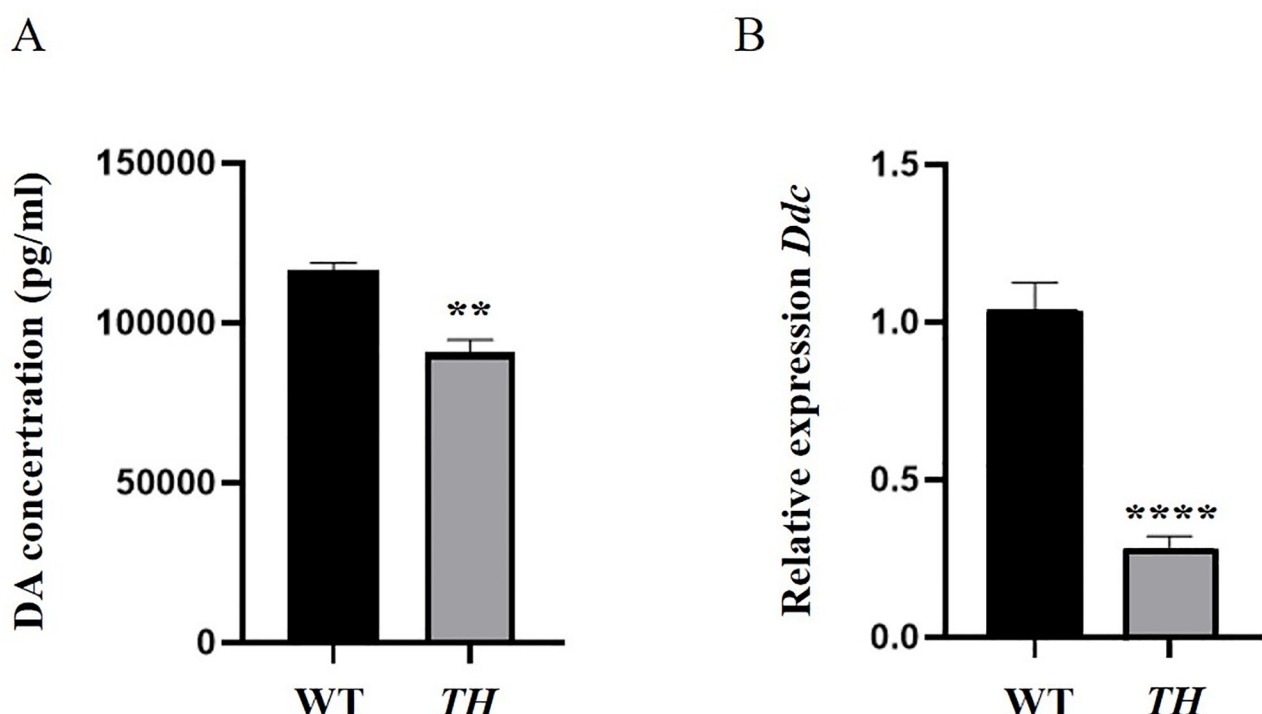

**Fig 11. Effects of *TH* knockout on dopamine concentrations (A) and the expression of cuticle tanning pathway gene: Dopa decarboxylase (*Ddc*)** **(B) P < 0.0001.** ($t$ = 7.361, $df$ = 22) Three replications are conducted, and the data are presented as mean ± SE. Significant differences between the two treatments are analyzed using a Student's t-test.

bodies are also defective, such as incomplete molting and defective wings and legs. We tried to cross with WT to obtain $F_1$ heterozygotes, and the incubation rate of $F_1$ was significantly reduced. The heterozygotes of $F_1$ generation successfully incubated were normal in phenotype. But all the $F_2$ homozygotes died from molting. Therefore, the *TH* gene is essential for the growth and development of *G. bimaculatus*. The *yellow-y* gene is functionally located downstream of the *TH* gene [40]. In *Drosophila*, the *yellow-y* gene is highly expressed in the adult epidermis that produces melanin [22, 41, 42]. In *G. bimaculatus*, the body color is brown after the *yellow-y* gene knocking out. Germline genetic experiments fully demonstrate that the genes knocked out by CRISPR/Cas9 editing system can be inherited. In $F_1$ generation of *yellow-y* gene, the number ratio of WT to mutant was close to 1:1 (P = 0.026), so it can be preliminarily inferred that *yellow-y* gene was located on the autochromosome. However, it is necessary to obtain homozygotes to verify this problem and further verify the function of related genes.

Feeding with *TH* dsRNA decreased the transcription level of target genes and feeding level of larvae, leading to larvae death *Plutella xylostella* [43]. In addition, in the study of *TH* gene function mediated by CRISPR/Cas9 gene editing technology in *Agrotis ipsilon*, the expression level of *TH* gene in the mutants was significantly decreased. The CRISPR/Cas9 system works at the genomic level, disrupting gene function more than RNAi inhibits gene expression at the transcriptional level. *TH* genes show different spatio-temporal expression patterns in different species, which may be closely related to the upstream and downstream regulatory elements [35, 44]. *TH* is involved in the first step of DOPA production from tyrosine [35, 45]. In *Tenebrio molitor* and *G. bimaculatus*, the *Ddc* gene is located downstream of the *TH* gene [39, 46]. Similar body color phenotype was obtained from RNAi silencing *TH* gene and *Ddc* gene expression levels in *G. bimaculatus*. In this study we found that the expression level of *Ddc* gene was significantly down-regulated in *TH* mutants of 1st-instar nymphs. The directly interaction of the two genes in *G. bimaculatus* still need further verification. In the follow-up experiment, we will detect the expression levels of *TH* and *Ddc* genes at different instars.

In this study, we use CRISPR/Cas9 system to knock out *TH* gene and *yellow-y* gene in *G. bimaculatus*. Significant phenotypes are observed in $F_0$ mosaic mutants. A phenotype similar to $F_0$ crispants is observed in the $F_1$ heterozygotes of the *yellow-y* gene with some variation. Because *TH* gene homozygous is lethal, it not only affects the pigmentation of *G. bimaculatus*, but also may affect the overall development and growth of *G. bimaculatus*. By now, this complex process and the regulatory signaling pathway needs further elucidate. In subsequent experiments, we will evaluate the causes of *TH* gene damage to development, focusing on the gene network and molecular interaction of *TH* genes and the earlier stage. We will also conduct further research on gene functions in the melanin synthesis pathway.

## Supporting information

**S1 Table. Species for the *TH* gene used in this study with GenBank accession numbers.** (DOCX)

**S2 Table. Species for the *yellow-y* gene used in this study with GenBank accession numbers.** (DOCX)

**S3 Table. Survival rate and percentage of mosaicism in $F_0$ crickets injected with *TH* and *yellow-y* sgRNAs.** (DOCX)

**S1 Fig. Phylogenetic trees are based on the amino acid sequences of 23 and 7 species, respectively.** (A) Phylogenetic tree of TH protein sequences. (B) Phylogenetic tree of yellow-y

amino acid sequences.
(TIF)

**S2 Fig. DNA sequencing results of WT eggs and eggs injected with *yellow-y* sgRNA and Cas9 protein.** The red box refers to the PAM sequence.
(TIF)

**S3 Fig. Phenotype and sequence analysis of *TH* gene mutants.** The first strand is the wild type, and the second strand is the mutant. Bar = 3mm.
(TIF)

**S4 Fig.**
(TIF)

**S1 Data.**
(XLSX)

## Acknowledgments

We gratefully thank Dr. Hun-Lun Bi and Xia Xu for providing constructive comments on the manuscript.

## Author Contributions

**Data curation:** Yun Bai, Yuan He, Dong-Liang Li, Zhu-Qing He.

**Funding acquisition:** Kai Li.

**Methodology:** Chu-Ze Shen, Kai Li, Dong-Liang Li.

**Writing – original draft:** Yun Bai.

**Writing – review & editing:** Dong-Liang Li, Zhu-Qing He.

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
