## [Decision Letter · Decision Letter 0]

23 Mar 2022

PONE-D-22-04580CRISPR/Cas9-Mediated Genomic Knock out of Tyrosine Hydroxylase and Yellow genes in Cricket Gryllus bimaculatusPLOS ONE

Dear Dr. He,

Thank you for submitting your manuscript to PLOS ONE. After careful consideration, we feel that it has merit but does not fully meet PLOS ONE’s publication criteria as it currently stands. Therefore, we invite you to submit a revised version of the manuscript that addresses the points raised during the review process. Your manuscript was reviewed by three experts. They found that it is preliminary and requires a major revision(s) including additional experiments. I share their recommendations. Please revise it according to their suggestions and perform additional experiments as they suggested.

We look forward to receiving your revised manuscript.

Kind regards,

Hodaka Fujii, M.D., Ph.D.

Academic Editor

PLOS ONE

Journal Requirements:

[This research was funded by the National Natural Science Foundation of China (No. 31801997) and the Natural Science Foundation of Shanghai (19ZR1416100). We also gratefully thank Dr. Hun-Lun Bi and Xia Xu for providing constructive comments on the manuscript.]

 [National Natural Science Foundation of China (No. 31801997)

Natural Science Foundation of Shanghai (19ZR1416100)

NO - Include this sentence at the end of your statement: The funders had no role in study design, data collection and analysis, decision to publish, or preparation of the manuscript.]

4. PLOS requires an ORCID iD for the corresponding author in Editorial Manager on papers submitted after December 6th, 2016. Please ensure that you have an ORCID iD and that it is validated in Editorial Manager. To do this, go to ‘Update my Information’ (in the upper left-hand corner of the main menu), and click on the Fetch/Validate link next to the ORCID field. This will take you to the ORCID site and allow you to create a new iD or authenticate a pre-existing iD in Editorial Manager. Please see the following video for instructions on linking an ORCID iD to your Editorial Manager account: https://www.youtube.com/watch?v=_xcclfuvtxQ.

5. Please ensure that you refer to Figures 6, 7 and 8  in your text as, if accepted, production will need this reference to link the reader to the figures.

6. Please include captions for figures 6, 7 and 8.

7. We note you have included a table to which you do not refer in the text of your manuscript. Please ensure that you refer to Table 3 in your text; if accepted, production will need this reference to link the reader to the Table.

Additional Editor Comments:

Please correct the following wordings:

Abstract, p. 2, line 28: "an efficient transient CRISPR/Cas9 system" to "an efficient transient CRISPR/Cas9 expression system"

Results, p. 8, line 166: "Fig 2" might be a mistake. Please cite a correct figure.

Results, p. 9, line 179: "Fig 3" might be a mistake. Please cite a correct figure.

Reviewers' comments:

Reviewer's Responses to Questions

**Comments to the Author**

1. Is the manuscript technically sound, and do the data support the conclusions?

Reviewer #1: Partly

Reviewer #2: Partly

Reviewer #3: Partly

2. Has the statistical analysis been performed appropriately and rigorously? 

Reviewer #1: N/A

Reviewer #2: N/A

Reviewer #3: I Don't Know

3. Have the authors made all data underlying the findings in their manuscript fully available?

Reviewer #1: No

Reviewer #2: Yes

Reviewer #3: Yes

4. Is the manuscript presented in an intelligible fashion and written in standard English?

Reviewer #1: Yes

Reviewer #2: No

Reviewer #3: No

5. Review Comments to the Author

Reviewer #1: Thank you for giving me the opportunity to review “CRISPR/Cas9-Mediated Genomic Knock out of Tyrosine Hydroxylase and Yellow genes in Cricket Gryllus bimaculatus” by Bai et al.

In this manuscript, the authors attempted genome editing using CRISPR/Cas9 system into the cricket, Gryllus bimaculatus. Then, they have successfully inducing gene knock-out mutation against TH and yellow-y genes. This genome editing method will not only benefit the cricket research community but will also be important in studies among insect species, especially for researchers who use hemimetabola insects.

However, while some potentially interesting results have been observed in this study, there are some concerns. The conclusion of this manuscript is that established the genome modification in G. bimaculatus. But it limits the novelty of this study because of already been published in other groups. Therefore, if the authors more emphasize the functions of melanin pigmentation-related genes, it will be more attractive to broad readers. This is because, from this reviewer's knowledge, findings from studies on the function of cuticular coloration in hemimetabola insects, including cricket, are not fully understood.

Major concerns:

The text is organized but there is missing information and thus the data should be incorporated in the manuscript with referred following comments.

The main concern to this manuscript is that the authors did not focus on expression levels of TH and yellow-y genes between wt and those mutants. Therefore, it is hard to understand the mutant phenotypes are due to null or hypomorphic mutations. It should be shown that the expression level of TH and yellow-y genes were decreased or not on the mutant by (semi-) quantitative PCR. Otherwise, at least it should be compared the phenotypes by those RNAi mutants.

Line 176 at the section of Phenotypic analysis: Please more describe details in TH and yellow-y mutants with showing dissected parts or with enlargement of the figure for the parts. In holometabola insects, the cuticular coloration pattern differs among tissues (i.e., the compound eye and the majority of body cuticles). These phenotypic differences could be observed in results (Fig. 3 and Fig. 4).

Is there any evidence the Gryllus yellow-y used in this experiment is certainly the gene classified in yellow-y among insects, beyond the blast search? Please take considering phylogenetic analysis using Yellow family proteins among insects including Gryllus yellow genes.

The melanization in insect pigmentation undergoes enzymatic cascades. It is possible that mutants with weakened enzymatic activity may be slower than normal in coloration. Please show the time course changes in body coloration of the wild type and the mutant during development.

Please prepare the table having injection statistical numbers. The table should have at least numbers of Injected eggs, Hatched nymphs, Visible mosaic adults. Adding for yellow-y, it should have the number of F0 adults crossed and F1 with color mutation.

Minor concerns:

Fig. 1 and 2, It is hard to understand where gRNA is bound in present figures in which the sequences were shown by amino acids. It should be shown gRNA location with the sequence of nucleic acids.

Fig. 1 and 2, Is there any functional domain in those genes? Please show overlaying the domains into current figures.

Line 105, Please describe what type of PCR enzyme was used for producing gRNA in this experiment.

Lin 219, Please use Holometabola and Hemimetabola to specify the metamorphic type.

Figure 3, Please center the PAM sequence in the sequence wave. This will give information on the mutation status of the surrounding 5’ and 3’ sequences.

Reviewer #2: The manuscript by Yun Bai et al., entitled “CRISPR/Cas9-Mediated Genomic Knock out of Tyrosine Hydroxylase and Yellow genes in Cricket Gryllus bimaculatus”, addresses the methods of gene knock-out using CRISPR/Cas9 system, targeting two pigmentation genes TH and yellow-y.

The Authors obtained TH knock-out F0, and yellow-y knock-out F0 and F1, using single gRNA and Cas9 protein, and showed their edited sequences and pigmentation defect phenotypes. In Gryllus bimaculatus, gene knock-out and targeted gene insertion mediated by gRNA and Cas9 mRNA are established. The authors claimed that genome editing using Cas9 protein is the preferred choice in Gryllus bimaculatus, like in Pyrrhocoris apterus, based on their results. However, genome editing method using Cas9 protein in G. bimaculatus is already established (Ohde et al., 2022), and authors did not fully perform to confirm their genome editing results. Thus, the novelty of their findings is quite limited.

Major comments

In figure 1, genome structures of TH and yellow-y genes are incorrect. TH gene (BAM15632.1) and yellow-y gene (GBI_10058-RA) contain at least 11 and 4 exons, respectively, in G. bimaculatus genome.

Line 166 and Figure 2A,B. The authors briefly describe the results of in vitro Cas9 cleavage assay, but the authors should describe their results more carefully. There are no information about the size of DNA marker. In figure 2A, the amounts of DNA samples are different in each lane; the amounts of DNA in lanes 2 and 3 looks much larger than in lanes 3 and 4. I think Cas9 cleavage assay should be done to same amount of DNA. In figure 2B, band patterns of lanes 2-7 were faint and out-of-focus, thus, it was difficult to distinguish the cleavage pattern. The authors should perform in vitro cleavage assay using same amounts of DNA fragments and show the lengths of DNA fragments or DNA marker.

lines 173-175. The authors claimed that DNA sequences of injected group showed multiple peaks, in figure 2B and C. The authors showed WT and TH in complimentary, but should show normal directions. The authors need to add the explanation about the red rectangles (maybe those indicate PAM sequences?).

About the phenotypic analysis. The authors showed TH and yellow-y knocking-out F0 nymphs in figure 3. In my knowledge, cricket exoskeleton is formed several layers and some layers accumulate melanin pigment, but other layers does not. The authors focused on the pigmentation genes functions in this paper, the authors should show the pigmentation defects in the exoskeleton by sectioning WT and injected F0 nymphs. Such experiments would provide the data related to the mosaicism of genome editing F0 nymphs using CRISPR/Cas9 system.

After melanin synthesis gene knocking-out, however, crickets showed brownish body color. The authors should show the results, or discuss, about other pigments, especially about ommochromes, which appears brownish or reddish color.

The authors claimed that most of TH genome editing F0 nymphs died before or during the first molting, but the authors did not mention about the reasons of their nymphal lethal phenotype. TH catalyses tyrosine to dopa, and dopa is not a substrate of melanin but also dopamine. I recommend that the authors should detect and compare the amount of dopamine in WT and TH genome editing nymphs, because dopamin is involved neuronal activeties.

The authors showed the photos of TH or yellow-y genome editing nymphs that appear mosaicisms in figure 3, and the authors determined the specific mutation types from these genome editing nymphs and showed the sequences in figure 5. However, the authors did not specify which sequences were obtained from which CRISPANTs. In figure 5, some sequences have 3, 6, 21 bp deletion or 9 bp insertion that leads in-frame mutation, thus, mutated TH or yellow-y protein have 1, 2, 7 amino acids deletion or 3 amino acids insertion. Generally, these small in-frame deletion or insertion may not affect an enzymatic activity. In figure 5A, TH genome editing nymph in the most left panel shows blackish pigment pattern, similar to the WT or Control, speculating that these blackish nymphs have in-frame mutation. The authors should carefully show the relationship between mutation patterns of TH and yellow-y genes and mutant phenotypes of pigmentation.

In Table 2 and Figure 5B, the authors showed phenotypes of yellow-y / WT F1 nymphs. The authors showed a ratio of mutant and WT as nearly 50% and 50% in both yellow-y male / WT female and yellow-y female / WT male. Is this ratios mean that genetic locus of yellow-y is on autosome, not on a sex chromosome? The authors should discuss about the ratio of F1 generation.

In figure 5D, the authors showed DNA sequencing results of TH and yellow-y genome editing eggs and WT. Again, the authors showed the sequences of yellow-y in complementary, but should be shown in normal direction. In the figure legends, the authors wrote that the red rectangles in the figure are the PAM sequences. In figure 5D, four letters CTCG in TH and GGTG in yellow-y were boxed in red rectangles, but the PAM sequences of Cas9 is three letters (NGG), not four letters. The authors should show the correct PAM sequences.

In general, expressions of the genome-edited genes (in this case, TH and yellow-y) of CRISPANTs and WT should be determined and compared in mRNA level and protein level, by using qPCR (or northern blot) and Western blot, in addition to the determination of DNA sequencing of targeted sites.

As the author mentioned in Discussion section, the authors should determine whether mutations occur in off-targets or not.

Minor comments

line 66 (so called "insertions") should be (so called "indels").

Line 122 "The vitro" should be "The in vitro".

Line 150 "TM" in "AxyPrep TM Multisource Genomic DNA Minipres Kit" should be in superscript.

Line 269 "Lim, K.-T.J.P.o." should be "Lim, K.-T.".

Line 282 ""Noji, S.J.S." should be "Noji, S.".

Line 295 The authors names and journal name are incorrect.

Lines 351-354 The authors names are incorrect.

Line 388 and 389-390 G. bimaculatus should be in italic.

Line 388 and 390 "5nd" should be "5th".

Line 395 "8nd" should be "8th".

Reviewer #3: In this paper, the authors examine function of TH and yellow genes in a model insect species, Gryllus bimaculatus, in the melanin pathway. They use CRISPR to target these genes and demonstrate mutant phenotypes. The results suggest that the mutagenesis using Cas9 protein rather than Cas9 mRNA is feasible for this species, and that mutagenesis for both TH and yellow genes causes body color chages in nymphs. The conculsion of this paper seems reasonable; however, the data presented in this study are rather preliminary and I have concerns about the quality of the data presented in the paper and the description of the results.

Overall, I feel that the paper is too preliminary to be considered for publication.

Major comments:

1) Although this paper emphasizes the success of mutagenesis using the Cas9 protein as an achievement, the use of the Cas9 protein is already a widely known method and is not novel, even if it is the first report of its use in crickets. Even leaving novelty aside, if the efficacy of Cas9 protein is to be emphasized, at least comparative data with the use of mRNA against the same target are needed.

2) Result page9：It is not appropriate to argue the function of TH from the phenotype at F0 described in this paper. There is a lack of analytical data on the phenotype (e.g., effect on cuticle thickness). In particular, although it is reasonable to assume that the lack of dopamine synthesis does indeed affect body color, it is difficult to determine from the results obtained whether this is a systemic or localized function. Although it is stated that most TH knockouts are lethal, not all, and it is considered possible to obtain the next generation by reducing the amount of Cas9 introduced, etc. TH gene function should be considered based on the phenotype of the heterozygous or homozygous knockout.

3) Line 183, Fig.4: The F1 phenotype of the Yellow gene knockout is described as brown or black body color. However, the correspondence between the described phenotype and genotype is unclear in the paper. It is essential to clarify the exact correspondence with the genotype in order to infer the gene function from the phenotype of the mutant.

4) The insect yellow gene family generally contains a large number of genes (as the authors also state in line 231 of the text). More detailed verification and description of which genes of the gene family have been knocked out is needed to reach the conclusions of this paper. Can the gene targeted by the authors be considered a true ortholog of the Drosophila yellow gene? The specificity of genome editing must also be verified, as genes of the same family are likely to have similar nucleotide sequences.

Minor comments:

5) Fig. 1, 2: For each of the TH and Yellow proteins, the functional domains should be shown and the targets of the guide RNA should be indicated.

6) All of the individuals showing the phenotypes shown in Figs. 3 and 4 appear to have abnormal antennae. Is the effect on the antennae significant?

7) Method: The authors should specify the database used to identify the TH and yellow genes in crickets.

8) line 225 : I could not understand the meaning of this sentence.

9) line 230-231: In the results section, there appears to be no description of the body defect due to TH gene knockout mentioned here. Also, what specific data is the statement that it is essential for "growth and development" based on? At least lethality does not necessarily mean that it is involved in growth and development.

10) Table1: This table is difficult to comprehend. What do the numbers in "gonadal mosaics" mean? Does it mean the efficiency of introducing mutations into the germplasm? It is also unclear what the numbers in Number of mutations mean. An explanation of how these numbers were calculated is needed.

6. PLOS authors have the option to publish the peer review history of their article (what does this mean?). If published, this will include your full peer review and any attached files.

Reviewer #1: No

Reviewer #2: No

Reviewer #3: No

---

## [Author Response · Author response to Decision Letter 0]

11 Aug 2022

Editor:

Thank you for submitting your manuscript to PLOS ONE. After careful consideration, we feel that it has merit but does not fully meet PLOS ONE’s publication criteria as it currently stands. Therefore, we invite you to submit a revised version of the manuscript that addresses the points raised during the review process.

Your manuscript was reviewed by three experts. They found that it is preliminary and requires a major revision(s) including additional experiments. I share their recommendations. Please revise it according to their suggestions and perform additional experiments as they suggested.

We would like to thank you first for all the suggestions of our manuscript (Manuscript Number: PONE-D-22-04580) entitled “CRISPR/Cas9-Mediated Genomic Knock out of Tyrosine Hydroxylase and Yellow genes in Cricket Gryllus bimaculatus”. We really appreciate your help and patience. We have seriously thought about the suggestions and provided our response to reviewers. 

We added an experiment to observe the pigmentation process of Gryllus bimaculatu (Fig. 6), and found that the WT group completed pigmentation within 1.5h, while the mutant completed pigmentation within 24 hours. The head of the mutant is mosaic compared with the WT (Fig. 7). Dopamine concentration (Fig. 11) measurement experiments showed that knocking out the TH gene resulted in a significant decrease. In addition, the phylogenetic analysis of TH and yellow-y genes was added (Fig. 2). The QRT-PCR experiment (Fig. 9) and the melanin content experiment (Fig. 10) also more fully verified that the content of melanin in the mutant group was significantly reduced.

The detailed response attached below.

Please ensure that you refer to Figures 6, 7 and 8 in your text as, if accepted, production will need this reference to link the reader to the figures. Please include captions for figures 6, 7 and 8.

We thank the reviewer for these suggestions. The references and correspondence for Figures 6, 7 and 8 have been rearranged as shown in Figure 8.

We note you have included a table to which you do not refer in the text of your manuscript. Please ensure that you refer to Table 3 in your text; if accepted, production will need this reference to link the reader to the Table.

Thank you for your advice. In this article, we have modified Table 3 on page 7, line 134 to Table 1.

Please correct the following wordings:

A: Abstract, p. 2, line 28: "an efficient transient CRISPR/Cas9 system" to "an efficient transient CRISPR/Cas9 expression system"

B: Results, p. 8, line 166: "Fig 2" might be a mistake. Please cite a correct figure.

C: Results, p. 9, line 179: "Fig 3" might be a mistake. Please cite a correct figure.

We thank the reviewer for these suggestions.

A: This has been modified, see line 28 on page 2.

B: The correct image has been referenced, as shown in Figure 2.

C: The correct image has been referenced, as shown in Figure 3.

Reviewer #1:

Thank you for giving me the opportunity to review “CRISPR/Cas9-Mediated Genomic Knock out of Tyrosine Hydroxylase and Yellow genes in Cricket Gryllus bimaculatus” by Bai et al.

In this manuscript, the authors attempted genome editing using CRISPR/Cas9 system into the cricket, Gryllus bimaculatus. Then, they have successfully inducing gene knock-out mutation against TH and yellow-y genes. This genome editing method will not only benefit the cricket research community but will also be important in studies among insect species, especially for researchers who use hemimetabola insects.

However, while some potentially interesting results have been observed in this study, there are some concerns. The conclusion of this manuscript is that established the genome modification in G. bimaculatus. But it limits the novelty of this study because of already been published in other groups. Therefore, if the authors more emphasize the functions of melanin pigmentation-related genes, it will be more attractive to broad readers. This is because, from this reviewer's knowledge, findings from studies on the function of cuticular coloration in hemimetabola insects, including cricket, are not fully understood.

Major concerns:

The text is organized but there is missing information and thus the data should be incorporated in the manuscript with referred following comments.

1. The main concern to this manuscript is that the authors did not focus on expression levels of TH and yellow-y genes between wt and those mutants. Therefore, it is hard to understand the mutant phenotypes are due to null or hypomorphic mutations. It should be shown that the expression level of TH and yellow-y genes were decreased or not on the mutant by (semi-) quantitative PCR. Otherwise, at least it should be compared the phenotypes by those RNAi mutants.

Thank you very much for your suggestion, we have added this part of the experiment as shown in Figure 9.

2. Line 176 at the section of Phenotypic analysis: Please more describe details in TH and yellow-y mutants with showing dissected parts or with enlargement of the figure for the parts. In holometabola insects, the cuticular coloration pattern differs among tissues (i.e., the compound eye and the majority of body cuticles). These phenotypic differences could be observed in results (Fig. 3 and Fig. 4).

Thank you for your reminding. We have added a comparison experiment between mutant and wild-type heads, see Figure 7.

3. Is there any evidence the Gryllus yellow-y used in this experiment is certainly the gene classified in yellow-y among insects, beyond the blast search? Please take considering phylogenetic analysis using Yellow family proteins among insects including Gryllus yellow genes. 

Thank you very much for your suggestion, and the yellow gene is a large family that we have mapped through phylogenetic analysis, see Figure 2.

4. The melanization in insect pigmentation undergoes enzymatic cascades. It is possible that mutants with weakened enzymatic activity may be slower than normal in coloration. Please show the time course changes in body coloration of the wild type and the mutant during development.

Thank you for your advice. We have added experiments to observe the process of body color changes in mutants and wild-types in Figure 6.

5. Please prepare the table having injection statistical numbers. The table should have at least numbers of Injected eggs, Hatched nymphs, Visible mosaic adults. Adding for yellow-y, it should have the number of F0 adults crossed and F1 with color mutation.

Thank you for your suggestion. We have explained the injection of yellow-y mutants by adding Tables 3 and 4.

Minor concerns:

6. Fig. 1 and 2, It is hard to understand where gRNA is bound in present figures in which the sequences were shown by amino acids. It should be shown gRNA location with the sequence of nucleic acids.

Fig. 1 and 2, Is there any functional domain in those genes? Please show overlaying the domains into current figures.

Thank you very much for your suggestion, and we have re-experimented and plotted the graphs validated in vitro, see Figure 3.

7. Line 105, Please describe what type of PCR enzyme was used for producing gRNA in this experiment.

Thank you for your suggestion. This has been added. See page 6, line 121.

8. Lin 219, Please use Holometabola and Hemimetabola to specify the metamorphic type.

Thank you for your suggestion. This has been modified. See page 19, line 347-348.

9. Figure 3, Please center the PAM sequence in the sequence wave. This will give information on the mutation status of the surrounding 5’ and 3’ sequences.

Thank you for your advice. We have modified the gel map and sequence orientation, see Figure 3.

Reviewer #2:

The manuscript by Yun Bai et al., entitled “CRISPR/Cas9-Mediated Genomic Knock out of Tyrosine Hydroxylase and Yellow genes in Cricket Gryllus bimaculatus”, addresses the methods of gene knock-out using CRISPR/Cas9 system, targeting two pigmentation genes TH and yellow-y.

The Authors obtained TH knock-out F0, and yellow-y knock-out F0 and F1, using single gRNA and Cas9 protein, and showed their edited sequences and pigmentation defect phenotypes. In Gryllus bimaculatus, gene knock-out and targeted gene insertion mediated by gRNA and Cas9 mRNA are established. The authors claimed that genome editing using Cas9 protein is the preferred choice in Gryllus bimaculatus, like in Pyrrhocoris apterus, based on their results. However, genome editing method using Cas9 protein in G. bimaculatus is already established (Ohde et al., 2022), and authors did not fully perform to confirm their genome editing results. Thus, the novelty of their findings is quite limited.

Major comments

In figure 1, genome structures of TH and yellow-y genes are incorrect. TH gene (BAM15632.1) and yellow-y gene (GBI_10058-RA) contain at least 11 and 4 exons, respectively, in G. bimaculatus genome.

Thank you for your suggestion. The exons of the TH and yellow-y genes of Gryllus bimaculatus have been modified by reviewing the transcriptome data and alignment with Drosophila sequences, see Figure 1.

Line 166 and Figure 2 A, B. The authors briefly describe the results of in vitro Cas9 cleavage assay, but the authors should describe their results more carefully. There are no information about the size of DNA marker. In figure 2 A, the amounts of DNA samples are different in each lane; the amounts of DNA in lanes 2 and 3 looks much larger than in lanes 3 and 4. I think Cas9 cleavage assay should be done to same amount of DNA. In figure 2B, band patterns of lanes 2-7 were faint and out-of-focus, thus, it was difficult to distinguish the cleavage pattern. The authors should perform in vitro cleavage assay using same amounts of DNA fragments and show the lengths of DNA fragments or DNA marker.

Thank you very much for your suggestion, and we have re-experimented and plotted the graphs validated in vitro, see Figure 3.

lines 173-175. The authors claimed that DNA sequences of injected group showed multiple peaks, in figure 2B and C. The authors showed WT and TH in complimentary, but should show normal directions. The authors need to add the explanation about the red rectangles (maybe those indicate PAM sequences?).

Thank you very much for your suggestion, and we have re-experimented and plotted the graphs validated in vitro, see Figure 3. The red rectangle represents the PAM sequence, see page 13, line 223.

About the phenotypic analysis. The authors showed TH and yellow-y knocking-out F0 nymphs in figure 3. In my knowledge, cricket exoskeleton is formed several layers and some layers accumulate melanin pigment, but other layers does not. The authors focused on the pigmentation genes functions in this paper, the authors should show the pigmentation defects in the exoskeleton by sectioning WT and injected F0 nymphs. Such experiments would provide the data related to the mosaicism of genome editing F0 nymphs using CRISPR/Cas9 system.

After melanin synthesis gene knocking-out, however, crickets showed brownish body color. The authors should show the results, or discuss, about other pigments, especially about ommochromes, which appears brownish or reddish color.

The authors claimed that most of TH genome editing F0 nymphs died before or during the first molting, but the authors did not mention about the reasons of their nymphal lethal phenotype. TH catalyses tyrosine to dopa, and dopa is not a substrate of melanin but also dopamine. I recommend that the authors should detect and compare the amount of dopamine in WT and TH genome editing nymphs, because dopamin is involved neuronal activeties.

Thank you for your advice. The cuticle is the outermost part of the insect body, and its physical properties, for example elasticity, thickness, strength and stiffness, are key determinants in maintaining insect locomotion, mechanical support, body shape and normal development (Wan et al., 2016). As the insect exoskeleton, the cuticle is also the first protective barrier to defend against multifarious environmental stresses and prevents the penetration of external compounds, including pathogens, dehydration, physical injury and insecticides (Balabanidou et al., 2018). However, the hard insect exoskeleton impedes further growth during molting and metamorphosis. Therefore, cuticle digestion and biosynthesis must occur periodically in insects. It is well known that the newly synthesized insect cuticle is pale and soft, and must undergo a tanning process (sclerotization and pigmentation) over a specific period to allow normal insect growth and development (Gorman and Arakane., 2010). The level of TH in the insect integument can influence the degree of cuticle tanning in Manduca sexta (Gorman et al., 2007). In insects, body pigments are mainly synthesized in epidermal cells through a complex cascade of biochemical reactions. Body pigments are important for insect morphology and protection insects from exogenous physical injury (Andersen, 2010).

The color of insect compound eye is determined largely by the nature of pigments (Ichiki et al., 2007). Xanthommatin (brown pigment) and pteridine (red pigment) are two key forms of this pigment (page 18, line 332 and page 19, line 333-341).

See Figure 11 for the dopamine content in the mutant group and the control group.

Wan C, Hao Z, Feng X. Structures, properties, and energy-storage mechanisms of the semi-lunar process cuticles in locusts. Scientific reports, 2016, 6(1): 1-13.

Balabanidou V, Grigoraki L, Vontas J. Insect cuticle: a critical determinant of insecticide resistance. Current opinion in insect science, 2018, 27: 68-74.

Gorman M J, Arakane Y. Tyrosine hydroxylase is required for cuticle sclerotization and pigmentation in Tribolium castaneum. Insect biochemistry and molecular biology, 2010, 40(3): 267-273.

Gorman M J, An C, Kanost M R. Characterization of tyrosine hydroxylase from Manduca sexta. Insect biochemistry and molecular biology, 2007, 37(12): 1327-1337.

Andersen S O. Insect cuticular sclerotization: a review. Insect biochemistry and molecular biology, 2010, 40(3): 166-178.

Ichiki R, Nakahara Y, Kainoh Y, et al. Temperature‐sensitive eye colour mutation in the parasitoid fly Exorista japonica Townsend (Dipt.: Tachinidae). Journal of Applied Entomology, 2007, 131(4): 289-292.

The authors showed the photos of TH or yellow-y genome editing nymphs that appear mosaicisms in figure 3, and the authors determined the specific mutation types from these genome editing nymphs and showed the sequences in figure 5. However, the authors did not specify which sequences were obtained from which CRISPANTs. In figure 5, some sequences have 3, 6, 21 bp deletion or 9 bp insertion that leads in-frame mutation, thus, mutated TH or yellow-y protein have 1, 2, 7 amino acids deletion or 3 amino acids insertion. Generally, these small in-frame deletion or insertion may not affect an enzymatic activity. In figure 5 A, TH genome editing nymph in the most left panel shows blackish pigment pattern, similar to the WT or Control, speculating that these blackish nymphs have in-frame mutation. The authors should carefully show the relationship between mutation patterns of TH and yellow-y genes and mutant phenotypes of pigmentation.

Thank you very much for your suggestion, and in the study of mutant phenotypes in this paper, individuals whose base pair insertion or deletion is not 3 times are selected, so these individuals all have frameshift mutations (page 10, line 165-166).

In Table 2 and Figure 5B, the authors showed phenotypes of yellow-y / WT F1 nymphs. The authors showed a ratio of mutant and WT as nearly 50% and 50% in both yellow-y male / WT female and yellow-y female / WT male. Is this ratios mean that genetic locus of yellow-y is on autosome, not on a sex chromosome? The authors should discuss about the ratio of F1 generation.

Thank you for your advice. This has been added to the discussion section, see lines 370-373 on page 20.

In figure 5 D, the authors showed DNA sequencing results of TH and yellow-y genome editing eggs and WT. Again, the authors showed the sequences of yellow-y in complementary, but should be shown in normal direction. In the figure legends, the authors wrote that the red rectangles in the figure are the PAM sequences. In figure 5 D, four letters CTCG in TH and GGTG in yellow-y were boxed in red rectangles, but the PAM sequences of Cas9 is three letters (NGG), not four letters. The authors should show the correct PAM sequences.

Thank you for your advice. The red boxes represent missing sequences. It has been modified on page 16, line 278.

In general, expressions of the genome-edited genes (in this case, TH and yellow-y) of CRISPANTs and WT should be determined and compared in mRNA level and protein level, by using qPCR (or northern blot) and Western blot, in addition to the determination of DNA sequencing of targeted sites.

Thank you very much for your suggestion, we have added this part of the experiment as shown in Figure 9.

As the author mentioned in Discussion section, the authors should determine whether mutations occur in off-targets or not.

Thank you for your advice. As mentioned earlier, there are many available online CRISPR design tools for predicting off-targets of particular sgRNAs (Cui et al., 2018). However, in our case, where information on the complete genome is unknown, the prediction of off-target effects by any of the design tool is impossible. Even with the draft of the genome available, where the manual search of off-targets (search against the occurrence of the “seed” region of the crRNA (Cho et al., 2014; Zhang et al., 2015) can be performed, we cannot exclude the possibility that some of the off-targets exist in unassembled repetitive regions (page 18, line328-331).

Cui Y, Xu J, Cheng M, et al. Review of CRISPR/Cas9 sgRNA design tools. Interdisciplinary Sciences: Computational Life Sciences, 2018, 10(2): 455-465.

Cho S W, Kim S, Kim Y, et al. Analysis of off-target effects of CRISPR/Cas-derived RNA-guided endonucleases and nickases. Genome research, 2014, 24(1): 132-141.

Zhang X H, Tee L Y, Wang X G, et al. Off-target effects in CRISPR/Cas9-mediated genome engineering. Molecular Therapy-Nucleic Acids, 2015, 4: e264.

Minor comments

line 66 (so called "insertions") should be (so called "indels").

Line 122 "The vitro" should be "The in vitro".

Line 150 "TM" in "AxyPrep TM Multisource Genomic DNA Minipres Kit" should be in superscript.

Line 269 "Lim, K.-T.J.P.o." should be "Lim, K.-T.".

Line 282 ""Noji, S.J.S." should be "Noji, S.".

Line 295 The authors names and journal name are incorrect.

Lines 351-354 The authors names are incorrect.

Line 388 and 389-390 G. bimaculatus should be in italic.

Line 388 and 390 "5nd" should be "5th".

Line 395 "8nd" should be "8th".

We thank the reviewer for these suggestions. We have revised the above problems in the article.

Reviewer #3:

In this paper, the authors examine function of TH and yellow genes in a model insect species, Gryllus bimaculatus, in the melanin pathway. They use CRISPR to target these genes and demonstrate mutant phenotypes. The results suggest that the mutagenesis using Cas9 protein rather than Cas9 mRNA is feasible for this species, and that mutagenesis for both TH and yellow genes causes body color chages in nymphs. The conculsion of this paper seems reasonable; however, the data presented in this study are rather preliminary and I have concerns about the quality of the data presented in the paper and the description of the results.

Overall, I feel that the paper is too preliminary to be considered for publication.

Major comments:

1) Although this paper emphasizes the success of mutagenesis using the Cas9 protein as an achievement, the use of the Cas9 protein is already a widely known method and is not novel, even if it is the first report of its use in crickets. Even leaving novelty aside, if the efficacy of Cas9 protein is to be emphasized, at least comparative data with the use of mRNA against the same target are needed.

Thank you for your advice. There are three primary sources of Cas9 delivery used for injections in genome editing experiments: (a) expression plasmid producing Cas9 mRNA and translated to protein in the host cell, (b) mRNA of Cas9, and (c) Cas9 protein (Bassett and Liu, 2014; Housden et al., 2014; Gratz et al., 2015; Kistler et al., 2015; Thurtle-Schmidt and Lo, 2018). In previous studies, further experiments were performed with a higher concentration of the Cas9 protein (500 ng/µl) in the injection mixture. Increased efficiency with the use of the Cas9 protein in the mutant generation is most probably caused by the higher stability in the host cells (Kotwica-Rolinska et al., 2019). And injecting the complex of sgRNA and Cas9 protein at the region of zygote formation allows Cas9 nuclease to edit genes immediately after injection (Hu et al., 2019). Both of Cas9 plasmid and mRNA could not function immediately (Kouranova et al. 2016); they need a certain time to experience a protein synthesis process in the embryo (Kouranova et al. 2016) and may miss the most suitable editing time (page 18, line 318-324).

Bassett A, Liu J L. CRISPR/Cas9 mediated genome engineering in Drosophila. Methods, 2014, 69(2): 128-136.

Housden B E, Lin S, Perrimon N. Cas9-Based Genome Editing in Drosophila. The Use of CRISPR/cas9, ZFNs, TALENs in Generating Site Specific Genome Alterations 546. 2014.

Gratz S J, Rubinstein C D, Harrison M M, et al. CRISPR‐Cas9 genome editing in Drosophila. Current protocols in molecular biology, 2015, 111(1): 31.2. 1-31.2. 20.

Kistler K E, Vosshall L B, Matthews B J. Genome engineering with CRISPR-Cas9 in the mosquito Aedes aegypti. Cell reports, 2015, 11(1): 51-60.

Thurtle‐Schmidt D M, Lo T W. Molecular biology at the cutting edge: a review on CRISPR/CAS9 gene editing for undergraduates. Biochemistry and molecular biology education, 2018, 46(2): 195-205.

Kotwica-Rolinska J, Chodakova L, Chvalova D, et al. CRISPR/Cas9 genome editing introduction and optimization in the non-model insect Pyrrhocoris apterus. Frontiers in physiology, 2019: 891.

Hu X F, Zhang B, Liao C H, et al. High-efficiency CRISPR/Cas9-mediated gene editing in honeybee (Apis mellifera) embryos. G3: Genes, Genomes, Genetics, 2019, 9(5): 1759-1766.

Kouranova E, Forbes K, Zhao G, et al. CRISPRs for optimal targeting: delivery of CRISPR components as DNA, RNA, and protein into cultured cells and single-cell embryos. Human gene therapy, 2016, 27(6): 464-475.

2) Result page9：It is not appropriate to argue the function of TH from the phenotype at F0 described in this paper. There is a lack of analytical data on the phenotype (e.g., effect on cuticle thickness). In particular, although it is reasonable to assume that the lack of dopamine synthesis does indeed affect body color, it is difficult to determine from the results obtained whether this is a systemic or localized function. Although it is stated that most TH knockouts are lethal, not all, and it is considered possible to obtain the next generation by reducing the amount of Cas9 introduced, etc. TH gene function should be considered based on the phenotype of the heterozygous or homozygous knockout.

Thank you for your advice. We have obtained more mutants by reducing the injection volume, and will follow-up by constructing pure and mutant mutants to study the function of TH and yellow-y genes and more detailed tissue localization.

3) Line 183, Fig.4: The F1 phenotype of the Yellow gene knockout is described as brown or black body color. However, the correspondence between the described phenotype and genotype is unclear in the paper. It is essential to clarify the exact correspondence with the genotype in order to infer the gene function from the phenotype of the mutant.

Thank you for your advice. All mutants for subsequent functional verification were selected from individuals whose number of mutated bases was not a multiple of 3. The sequence map of the number of missing bases presented in Figure 8 is to reflect the mutation efficiency.

4) The insect yellow gene family generally contains a large number of genes (as the authors also state in line 231 of the text). More detailed verification and description of which genes of the gene family have been knocked out is needed to reach the conclusions of this paper. Can the gene targeted by the authors be considered a true ortholog of the Drosophila yellow gene? The specificity of genome editing must also be verified, as genes of the same family are likely to have similar nucleotide sequences.

Thank you for your advice. We have performed a phylogenetic analysis of the yellow-y gene, see Figure 2.

Minor comments:

5) Fig. 1, 2: For each of the TH and Yellow proteins, the functional domains should be shown and the targets of the guide RNA should be indicated.

Thank you very much for your suggestion, and we have re-experimented and plotted the graphs validated in vitro, see Figure 3.

6) All of the individuals showing the phenotypes shown in Figs. 3 and 4 appear to have abnormal antennae. Is the effect on the antennae significant?

Thank you for your advice. Because the crickets were only slightly anaesthetized during the photo shoot, they also had a slow motion. So far, these genes have no effect on the development of crickets' antennae.

7) Method: The authors should specify the database used to identify the TH and yellow genes in crickets.

Thank you for your advice. In line 94-99 on page 5, database NCBI and InsectBase have been given.

8) line 225: I could not understand the meaning of this sentence.

Thank you for your advice. Lines 354-356 on page 20 have been revised.

9) line 230-231: In the results section, there appears to be no description of the body defect due to TH gene knockout mentioned here. Also, what specific data is the statement that it is essential for "growth and development" based on? At least lethality does not necessarily mean that it is involved in growth and development.

Thank you for your advice. This is explained on page 20, line 361-362.

10) Table1: This table is difficult to comprehend. What do the numbers in "gonadal mosaics" mean? Does it mean the efficiency of introducing mutations into the germplasm? It is also unclear what the numbers in Number of mutations mean. An explanation of how these numbers were calculated is needed.

We thank the reviewer for these suggestions. "Gonadal Mosaics" has been modified in Table 2. In addition, the data in the table are completed by statistical method of variance calculation for three repeated experimental data.

---

## [Decision Letter · Decision Letter 1]

31 Aug 2022

PONE-D-22-04580R1CRISPR/Cas9-Mediated Genomic Knock out of Tyrosine Hydroxylase and Yellow genes in Cricket Gryllus bimaculatusPLOS ONE

Dear Dr. He,

Thank you for submitting your manuscript to PLOS ONE. After careful consideration, we feel that it has merit but does not fully meet PLOS ONE’s publication criteria as it currently stands. Therefore, we invite you to submit a revised version of the manuscript that addresses the points raised during the review process.

Your revised manuscript was reviewed by the three reviewers who originally reviewed the original manuscript. Two of them still raised some concerns. Please revise it according to their suggestions. In this regard, a previous paper describing genome editing in cricket must be cited as one of them suggested. In addition, the scope of the manuscript should be shifted as they suggested. Furthermore, there are some inconsistencies in data (Figures and Tables). Please check them carefully and provide raw data for qPCR and others so that the reviewers can re-analyze them independently.

We look forward to receiving your revised manuscript.

Kind regards,

Hodaka Fujii, M.D., Ph.D.

Academic Editor

PLOS ONE

Reviewers' comments:

Reviewer's Responses to Questions

**Comments to the Author**

1. If the authors have adequately addressed your comments raised in a previous round of review and you feel that this manuscript is now acceptable for publication, you may indicate that here to bypass the “Comments to the Author” section, enter your conflict of interest statement in the “Confidential to Editor” section, and submit your "Accept" recommendation.

Reviewer #1: (No Response)

Reviewer #2: (No Response)

Reviewer #3: All comments have been addressed

2. Is the manuscript technically sound, and do the data support the conclusions?

Reviewer #1: Yes

Reviewer #2: No

Reviewer #3: Yes

3. Has the statistical analysis been performed appropriately and rigorously? 

Reviewer #1: N/A

Reviewer #2: No

Reviewer #3: Yes

4. Have the authors made all data underlying the findings in their manuscript fully available?

Reviewer #1: Yes

Reviewer #2: Yes

Reviewer #3: Yes

5. Is the manuscript presented in an intelligible fashion and written in standard English?

Reviewer #1: Yes

Reviewer #2: No

Reviewer #3: Yes

6. Review Comments to the Author

Reviewer #1: The authors have addressed most of the criticisms and performed several suggested experiments to address critical points. Although the authors responded well to the reviewers' comments several minor changes are still required.

I don’t know whether the journal accepted the style, but it makes it hard to read the figure legends embedded in the result section. The figure legends should be in a different section.

In Figure 1, please include UTRs and CDS to understand gene structures.

Line 216 and Figure 3. The author performed in vitro cleavage assay to evaluate gRNA efficiency. Could you please specify the size (length) of the target amplicon? Then also, please show the size of the cleaved amplicon.

Line 93 and Figure 1. Please organize the accession number of protein sequences in insects that are used in the phylogenetic analysis in a supplemental text.

Line 181. Please describe what tissues were used, sample number, and biological replicates leading to the statistical changes by the qPCR.

Reviewer #2: (No Response)

Reviewer #3: (No Response)

7. PLOS authors have the option to publish the peer review history of their article (what does this mean?). If published, this will include your full peer review and any attached files.

Reviewer #1: No

Reviewer #2: No

Reviewer #3: No

---

## [Author Response · Author response to Decision Letter 1]

15 Nov 2022

Dear Editor,

We would like to thank you first for all the suggestions of our manuscript (Manuscript Number: PONE-D-22-04580) entitled “CRISPR/Cas9-Mediated Genomic Knock out of Tyrosine Hydroxylase and Yellow genes in Cricket Gryllus bimaculatus”. We really appreciate your help and patience. We have seriously thought about the suggestions and provided our response to reviewers.

In Fig. 1, the structure of TH and yellow-y genes is analyzed in more details, for example, coding region and UTRs structure are supplemented. We show the deformity of leg and wing by TH gene knocking out of Gryllus bimaculatus by adding Fig. 5. We re-analyzed the qPCR data of TH, yellow-y and Ddc genes and normalized them, as shown in Fig. 10 and 12. We add references and related discussions on the interaction between TH and Ddc genes. TH genes show different spatio-temporal time and tissue expression patterns in different species, which may be closely related to the upstream and downstream regulatory elements (Gorman et al., 2007; Yu et al., 2011). In addition, we add supplementary materials (Supplementary Fig. 2) to explain the phenotypic and functional differences between the gene changes of multiples and non-multiples of three base-pairs. We illustrate the specific mutation types of TH and yellow-y genes in more details in the supplementary material Fig. 1 and 3.

Gorman M J, An C, Kanost M R. Characterization of tyrosine hydroxylase from Manduca sexta. Insect biochemistry and molecular biology, 2007, 37: 1327-1337. 

Yu H S, Shen Y H, Yuan G X, et al. Evidence of selection at melanin synthesis pathway loci during silkworm domestication. Molecular biology and evolution, 2011, 28: 1785-1799.

Reviewer #1: 

The authors have addressed most of the criticisms and performed several suggested experiments to address critical points. Although the authors responded well to the reviewers' comments several minor changes are still required.

I don’t know whether the journal accepted the style, but it makes it hard to read the figure legends embedded in the result section. The figure legends should be in a different section.

Thank you for your suggestion. We modified it.

In Figure 1, please include UTRs and CDS to understand gene structures.

We have modified it, as shown in new Fig. 1.

Line 216 and Figure 3. The author performed in vitro cleavage assay to evaluate gRNA efficiency. Could you please specify the size (length) of the target amplicon? Then also, please show the size of the cleaved amplicon.

Thank you for your advice. The target strip can be cut into two segments of different sizes. The original length of TH gene is 545bp, which is digested into 464bp. The original length of yellow-y gene is 1058bp and is digested into 826bp. We added these details in revision, please see lines 204 to 207 on page 12.

Line 93 and Figure 1. Please organize the accession number of protein sequences in insects that are used in the phylogenetic analysis in a supplemental text.

We revised it, please see supplementary material Table 1 and Table 2.

Line 181. Please describe what tissues were used, sample number, and biological replicates leading to the statistical changes by the qPCR.

Thank you for your advice. qPCR was performed to analyze the expression levels of TH and yellow-y genes with three 1st-instar nymphs of mutant and WT. All experiments were performed in three biological replicates. We have added these details in the text, see page 10, lines 168 to 173.

Reviewer #2:

The authors added new data and figures suggested by review comments. But the manuscript is still not well written especially in the Results section, and authors did not discuss about new data. The figures, graphs, and tables appear to be improperly prepared. Therefore, the authors need to rework the figures and graphs appropriately. The author claimed that main focus of this manuscript is establishment of genome editing strategy by using Cas9 protein, however, the strategy is already established from other laboratory (https://doi.org/10.1038/s41467-022-28624-x), as I mentioned in my previous review comments, although the authors did not cite this reference and discuss about the strategy. Therefore, the authors should shift the focus from establishment of genome editing strategy to molecular functions of pigmentation process.

We thank the reviewer for these very meaningful suggestions. When we wrote this manuscript, this research had not yet been published. In new revision, we refer to the new article on the construction of stable mutant strains of Gryllus bimaculatus by injecting Cas9 protein, see lines 265-267 on page 15 and see lines 304-305 on page 17. Following the advice, we will further study the gene function of body color-related genes and the underlying molecular mechanisms by achieving homozygous mutant strains. 

Major comments:

1. About abbreviation. The author uses abbreviations such as "WT," "CRISPR," "TH," "mRNA," "SE," and "Gryllus." Generally, when an abbreviation appears for the first time, it is given in full spelling, and the next time it appears, all abbreviations are given in abbreviated form. In this manuscript, the full spelling and abbreviations appear several times, making it difficult to read.

Thank you for your advice. All abbreviations have been checked and revised.

2. Line 210-212

The authors roughly wrote the results of in vitro verification, but the authors should show the results in detail. Original lengths of TH and yellow-y PCR products and predicted lengths after in vitro verification should be shown. Since there are no information on the sizes of PCR products and digested segments, it is impossible to determine if the experiments were successful or not.

Thank you for your advice. The target strip can be cut into two segments of different sizes. The original length of TH gene is 545bp, which is digested into 464bp. The original length of yellow-y gene is 1058bp and is digested into 826bp. We added these details in revision, please see lines 204 to 207 on page 12.

3. Line 213 and others

About figure legends. Please add a line break after the figure description, instead of continuing it with the main text.

Thanks. We have modified it.

4. Line 220-223 and Table 2

The authors wrote the percentages of mutant rate in main text and Table 2, but the values are different between them. I think the authors showed rounded values in Table 2, but the values should be in consistent.

In some case, the values are completely different between main text and table. For example, rounded value of "7.4%" in the main text is not "8%" in Table 2 for 300 ng/uL case. I am concerned about whether there is a data mix-up.

Thank you very much for your suggestion. The data in previous table was obtained by analysis of variance anova, while the data in the text was processed by means. This has now been carefully checked and changed to a uniform format. We corrected it in revision, see lines 208 to 214 on page 12. 

5. Line 224-225

The authors claimed that the mutant showed double peaks or random peaks at the target site. But in yellow-y mutant, only few double peaks are observed far from the PAM sequences. The authors should add the explanation about the direct sequence data of figure 3B and D. 

In addition, PAM sequences shown by red boxes in figure 3D is different from the PAM sequences shown by red texts in figure 8B, so it is impossible to determine which is the actual PAM sequence.

Thank you very much for your suggestions. Mutations in the gene were random at the target, and sequencing results close to PAM sequence mutations were added in the supplementary material Fig. 1. We have made further supplementary explanations to Fig. 3B and D, as shown in Supplementary Fig. 1. In addition, the annotations have been modified in Fig. 8B and D, and the annotation format of PAM has been unified. The red box represents the PAM sequence and the green box represents the missing base sequence.

6. Line 243-244, Line 254, and Figure 7

The authors wrote "The mutant heads of TH and yellow-y genes are mosaic-shaped (Fig7)" in line 243-244. In line 254, The authors wrote "(B) TH mutant HEAD pigment patterns. (C) The HEADS of yellow-y mutants are mosaic.", but in Figure 7, the authors showed TWO TH mutants heads and ONE yellow-y mutant head. The numbers of heads in Figure 7 and figure legend are different, so I could not understand which photo is TH mutant(s) and yellow-y mutant(s).

The second head In figure 7 seems bright and mosaic pigment patterns, but the third and fourth heads seems darker and uniformed pigment pattern. Thus, I think that the second head is TH mutant and the third and fourth head are yellow-y mutant, but it is still unclear why yellow-y mutant heads showed uniformed pigment pattern. The authors should present the generation of these mutant. I think TH mutant is F0 generation but yellow-y mutants could be F1. I am concerned that there may be a data mix-up.

Thank you for your advice. In Fig. 8, two TH mutant heads and one yellow-y mutant head are shown. The mutant heads of TH and yellow-y genes are mosaic-shaped, with the TH mutant having a light brown and black mosaic head, and the yellow-y mutant having a dark brown and black mosaic head. We have revised the description in the figure note, see lines 480 to 482 on page 33. The "head" in Fig. 8 is not plural, which caused misunderstood.

7. Lines 266-267, 270-271 and figure 8 The authors describe the results of DNA sequencing after TA cloning of TH and yellow-y mutants. The authors showed thirteen sequences of TH F0 and three and three sequences of yellow-y F0 and F1 in figure 8A and 8B, but explanation about sequencing results are too rough in line 270-271. Are there differences in pigment patterns between indels? For example, -21 bp, -6 bp, -3 bp, and +9 bp will not cause frame-shift mutation, thus mutated tyrosine hydroxylase would work and mutants containing these indels may show normal pigment pattern. Mutant nymphs containing other indels may cause frame-shift mutation and may produce non-functional enzyme, so these mutant nymphs may show defects in pigmentation. The authors should describe their experimental results in more detail.

In figure 8C and 8D, the authors showed wave data of TH mutant and yellow-y mutant in F1 generation. Both mutated sequences showed 4 bp deletion, but the authors should show the reason why they showed only these sequences.

The directions of nucleotide sequences and wave data shown in figure 8A, 8C and figure 3B are opposite. In addition, for TH gene case, the PAM sequences in figure 8A (the authors showed it in red) is different from the PAM sequences in figure 3B. For the yellow-y case, PAM sequences of figure 8B, 8D and figure 3D are different. Which is the correct PAM sequences? Where is the target site? The authors should correct figure 8 and figure 3, showing correct target sequences and PAM sequences.

Thank you very much for your suggestions.

In revision, we added the correlation between base pair deletion being a multiple of three or not a multiple of three and phenotype, and we also list some phenotypes, as shown in supplementary Fig. 2. The phenotype is clear when the number of missing bases is not a multiple of three.

Some representative results are selected for display in Fig. 8. We show more mutation types, such as missing 7bp and adding 1bp, in the supplementary materials Fig. 2 and Fig. 3.

The annotations have been modified in Fig. 8B and D, and the annotation format of PAM has been unified.

8. Line 277-278

The authors showed down-regulation of TH and yellow-y transcripts in these mutants. The authors designed target sites of sgRNA in coding region, not in regulatory elements of these genes, so why transcripts were decreased? I have often experienced such down-regulation of target gene's transcripts. The authors need to discuss about the down-regulation of the target genes citing appropriate references.

It was reported that the dietary TH double-stranded RNA led to reduced target gene transcript levels and larval feeding levels and caused larval mortality of diamondback moth (Ellango et al., 2018). In addition, in the study of TH gene function mediated by CRISPR/Cas9 gene editing technology in Agrotis ipsilon, it was also found that the expression level of TH gene in the mutants was also significantly decreased. Compared to RNA interference technology inhibiting gene expression at the transcription level, the CRISPR/Cas9 system works at genomic level, which is more destructive to gene function. TH genes show different spatio-temporal expression patterns in different species, which may be closely related to the upstream and downstream regulatory elements (Gorman et al., 2007; Yu et al., 2011). In Tenebrio molitor, the Ddc gene is located downstream of the TH gene (Mun et al., 2020). This part has been added in the article, please see page 18, lines 318-328.

Ellango R, Asokan R, Chandra G S, et al. Tyrosine hydroxylase, a potential target for the RNAi-mediated management of diamondback moth (Lepidoptera: Plutellidae). Florida Entomologist, 2018, 101: 1-5.

Gorman M J, An C, Kanost M R. Characterization of tyrosine hydroxylase from Manduca sexta. Insect biochemistry and molecular biology, 2007, 37: 1327-1337. 

Yu H S, Shen Y H, Yuan G X, et al. Evidence of selection at melanin synthesis pathway loci during silkworm domestication. Molecular biology and evolution, 2011, 28: 1785-1799.

Mun S, Noh M Y, Kramer K J, et al. Gene functions in adult cuticle pigmentation of the yellow mealworm, Tenebrio molitor. Insect biochemistry and molecular biology, 2020, 117: 103291.

9. Line 279-283

About figure 9. The authors showed relative expression levels of TH and yellow-y, and they set the mean of the relative expressions in WT to 1, but in figure 9, bars of TH and yellow-y of WT are slightly different from 1.0 of vertical axis. Especially in Figure 9A, gray bar of TH relative expression appears to be stretched vertically. The authors must rework the graph. The authors added asterisks to TH and yellow-y relative expressions, but there is no information about the numbers of asterisks. The authors should show the P values of student's t-test.

Thank you very much for your suggestion. We re-analyzed the qPCR data of TH, yellow-y and Ddc genes and normalized them, as shown in Fig. 10. Four stars represents a P value less than 0.0001.

10. Line 285

In Table 4, the authors showed the ratio of Mutant and WT as numbers of offspring and percentage. For yellow-y male x WT female case, sum of Mutant (58.5%) and WT (46.8%) percentages exceeds 100%. In general, sum of percentage never excess 100%. Are individuals with mosaic pigment patterns counted as both mutant and wild type? The authors need to explain why.

Thank you for your advice. This data is due to a calculation error and has been corrected in revision. The correct results should be 53.19% and 46.81%.

11. Line 290-291

The authors showed down-regulation of relative expression of Ddc in TH mutant in figure 11B. TH and Ddc are completely different genes, and both encoding enzymes, not coding transcription factors. The authors should explain why Ddc expression is down-regulated in TH mutants.

Thank you for your reminding. We have explained this in question 8. We have added discussion of the relationship between TH and Ddc gene interactions to the discussion section of the article, please see page 18, lines 318-328. We have modified it in Table 4.

12. Line 294-295

In figure 10C, the authors showed absorbance at 490 nm. Are the bars in this graph the mean and standard error? The authors should show the sample size (numbers of nymphs) and statistics. What is the meaning of asterisk on the gray bar?

Thank you for your advice. The bars in this graph the mean and standard error. Three 1st-instar nymphs of the same size were used in the experiment. A * indicates a P value less than 0.05, please see page 36, lines 501.

13. Line 297-300, figure 11

In line 297, the authors describe "three cuticle tanning pathway gene: dopamine decarboxylase". I think Ddc is a short gene name of "dopa decarboxylase", not "dopamine decarboxylase". And what is the meaning of "three"? Does Gryllus bimaculatus have three Ddc genes? The authors should add the information about "three" cuticle tanning pathway gene in Introduction or other appropriate sentences.

In figure 11B, the authors showed relative expression of Ddc in WT and Mutant. In this figure, did the authors set bar of WT as 1 or not? 

The authors should explain the meaning of four asterisks on the gray bar of Ddc relative expression in TH mutant.

On the vertical bar, the authors labeled as "GbDDC", but it should be "Ddc", in consistent with the main text.

Thank you for your reminding.

The word "Three" is a writing error and has been corrected in the article, please see cuticle tanning pathway gene page 37, lines 497-500.

We have performed a new analysis of the relative expression levels of Ddc genes, see Fig. 12. In addition, new additions have been made to the description of Fig. 12, see page 37, lines 497-500. In the data analysis, we have normalized the data again. Four stars represents a P value less than 0.0001.

In this article, we have changed "GbDDC" to "Ddc".

14. Line 316-317

The authors discussed about off-target effects. It is necessary to investigate whether there are off-target effects on the Ddc gene in TH mutants, at least, in relation to the down-regulation of Ddc expression.

Thank you for your advice. We have added discussion of the relationship between TH and Ddc gene interactions to the discussion section of the article, see page 18, lines 327-338. TH genes show different spatio-temporal expression patterns in different species, which may be closely related to the upstream and downstream regulatory elements (Gorman et al., 2007; Yu et al., 2011). In Tenebrio molitor, the Ddc gene is located downstream of the TH gene (Mun et al., 2020). 

Gorman M J, An C, Kanost M R. Characterization of tyrosine hydroxylase from Manduca sexta. Insect biochemistry and molecular biology, 2007, 37: 1327-1337. 

Yu H S, Shen Y H, Yuan G X, et al. Evidence of selection at melanin synthesis pathway loci during silkworm domestication. Molecular biology and evolution, 2011, 28: 1785-1799.

Mun S, Noh M Y, Kramer K J, et al. Gene functions in adult cuticle pigmentation of the yellow mealworm, Tenebrio molitor. Insect biochemistry and molecular biology, 2020, 117: 103291.

15. Line 342

The authors described that target site of TH is at the fifth exon. In line 95 and figure 1, The authors described target site is on the seventh exon. Which is correct?

Thank you for your reminding. That the target of the TH gene on the seventh exon is correct and has been modified in the article, see page 17, line 306.

16. Line 346

The authors discuss about the defective wings and legs, but no photo in figures or description in Result section. The authors should show the abnormal morphologies in the wings and legs.

Thank you for your advice. We have added the phenotype of leg and wing deformities in new Fig. 5.

Minor comments:

Line 95-96

I think the target site of TH gene in on the sixth exon. Please confirm the position.

By comparing the sequences of TH gene in the transcriptome database, we found that our target was the seventh exon of TH gene. We have also identified the sequence and the target by phylogenetic analysis.

Title

"and" should be non-italic in title "Tyrosine Hydroxylase and Yellow genes"

Lines 93-95

GBI_09528-RA is not an accession number of NCBI. The authors cite the reference written by Ylla et al, but the reference is for cricket genome sequence, not for transcriptome database.

Line 126

"sgRNA" in not mRNA.

Table 1

The authors should show the nucleotide sequences of qPCR primers for Ddc.

Line 199

"qRT-PCR" should be "qPCR".

Line 200-201

"Student'st-test "should be "Student's t-test".

Line 204

The authors should cite figure 1 in the sentence.

Line 206

"yellow" should be in italic.

Line 235

"dysfunction tyrosine yellow-y" would be "dysfunction yellow-y"

Line 237 and 240

The authors showed the percentages of mutant phenotype as "71.43%", "58.5%" and "55.6%". Does it make sense to have different decimal places represented? I feel that the decimal places should be aligned.

Line 265 and 266

"larvae" should be "nymphs"

Line 267

"wild egg" should be "wild type eggs"

Line 269

I don't know what "missing gene" means. Does it mean "missing nucleotides" ?

Line 280

The authors showed "TH" and "yellow-y" in the figure legend, but in figure 9, gene names are shown as ''GbTH" and "Gbyellow-y" at the vertical axes. These gene names should be "TH" and "yellow-y".

Line 281

"Aiactin" should be "beta-actin".

Line 288

"1-st instar" should "1st-instar".

Line 313

Add the appropriate references in "mRNA (ref)".

Line 334

"holometabola and hemimetabola" should be "holometabolous and hemimetabolous”.

We thank the reviewer for these suggestions. We have revised them all.

---

## [Decision Letter · Decision Letter 2]

6 Dec 2022

PONE-D-22-04580R2CRISPR/Cas9-Mediated Genomic Knock out of Tyrosine Hydroxylase and Yellow genes in Cricket Gryllus bimaculatusPLOS ONE

Dear Dr. He,

Thank you for submitting your manuscript to PLOS ONE. After careful consideration, we feel that it has merit but does not fully meet PLOS ONE’s publication criteria as it currently stands. Therefore, we invite you to submit a revised version of the manuscript that addresses the points raised during the review process.

 Your manuscript was reviewed by the three referees who had reviewed the original and revised manuscripts. As some reviewers pointed out, your manuscript needs an additional revision. Please revise your manuscript according to their suggestions. Especially,Please consider change the scope of your manuscript from method development to characterization of the pigmentation process as suggested by the reviewer 2.Please discuss more about the issues raised by the reviewer 2.Please carefully edit your manuscript to avoid any careless mistakes, typos, and grammatical errors. I would suggest to use a professional English editing service.Please submit your revised manuscript by Jan 20 2023 11:59PM. If you will need more time than this to complete your revisions, please reply to this message or contact the journal office at plosone@plos.org. Please include the following items when submitting your revised manuscript:A rebuttal letter that responds to each point raised by the academic editor and reviewer(s). You should upload this letter as a separate file labeled 'Response to Reviewers'.A marked-up copy of your manuscript that highlights changes made to the original version. You should upload this as a separate file labeled 'Revised Manuscript with Track Changes'.An unmarked version of your revised paper without tracked changes. You should upload this as a separate file labeled 'Manuscript'.If applicable, we recommend that you deposit your laboratory protocols in protocols.io to enhance the reproducibility of your results. Protocols.io assigns your protocol its own identifier (DOI) so that it can be cited independently in the future. For instructions see: https://journals.plos.org/plosone/s/submission-guidelines#loc-laboratory-protocols. Additionally, PLOS ONE offers an option for publishing peer-reviewed Lab Protocol articles, which describe protocols hosted on protocols.io. Read more information on sharing protocols at https://plos.org/protocols?utm_medium=editorial-email&utm_source=authorletters&utm_campaign=protocols.

We look forward to receiving your revised manuscript.

Kind regards,

Hodaka Fujii, M.D., Ph.D.

Academic Editor

PLOS ONE

Journal Requirements:

Reviewers' comments:

Reviewer's Responses to Questions

**Comments to the Author**

1. If the authors have adequately addressed your comments raised in a previous round of review and you feel that this manuscript is now acceptable for publication, you may indicate that here to bypass the “Comments to the Author” section, enter your conflict of interest statement in the “Confidential to Editor” section, and submit your "Accept" recommendation.

Reviewer #1: All comments have been addressed

Reviewer #2: (No Response)

Reviewer #3: All comments have been addressed

2. Is the manuscript technically sound, and do the data support the conclusions?

Reviewer #1: Yes

Reviewer #2: Partly

Reviewer #3: Yes

3. Has the statistical analysis been performed appropriately and rigorously? 

Reviewer #1: N/A

Reviewer #2: Yes

Reviewer #3: Yes

4. Have the authors made all data underlying the findings in their manuscript fully available?

Reviewer #1: Yes

Reviewer #2: Yes

Reviewer #3: Yes

5. Is the manuscript presented in an intelligible fashion and written in standard English?

Reviewer #1: Yes

Reviewer #2: No

Reviewer #3: Yes

6. Review Comments to the Author

Reviewer #1: Although the authors responded well to the reviewers’comments, several minor changes are still required.

In Table 1, Please describe the differences of the nucleotide shown capital and small letters.

Please match the decimal place that used in the main text and table.

In Figure 1, UTRs should be UTR.

Reviewer #2: The authors reworked Figures properly and modified Tables with appropriate values. The authors cited new references and added discussion, but the research focus of this manuscript was still to establish and optimize the genome editing strategy using Cas9 protein, and the authors did not shift to the research focus to pigmentation process. The genome editing strategy using Cas9 protein were already established as I mentioned through the review comments, so the novelty of this manuscript is none if the authors insist to publish the manuscript as methods paper. In addition, I recommended to add discussion about the Ddc gene expression in previous review comment, but was not enough discussed. I strongly recommend all authors must read and check all the text, figures and tables, because there are lots of careless mistakes in this manuscript.

Major comments:

1. The authors modified figure 3 and figure 9. In these figures, the authors showed that GGC sequences just adjacent to 5' side of sgRNA in yellow-y gene are the PAM sequences, but could be incorrect. PAM sequences must be NGG for Cas9 and locate just adjacent to 3' side of sgRNA, thus, I think that AGG sequences shown in figure 9D could be the PAM sequences. In figure 3D, I was confused that double peaks were appeared far from PAM sequences, as I pointed out in previous review comments. If the AGG was correct PAM sequences, double peaks in figure 3D were very close to PAM sequences and that is reasonable. The authors have to confirm the correct PAM sequences and should rewrite results and discussion and rework many figures, based on the position of PAM sequences.

2. To answer my previous comment, the authors discuss about the decreased expression of Ddc in TH knock out crickets in line 332-333, citing the genetic loci of Ddc and TH in Tenebrio molitor. However, in Gryllus bimaculatus, TH and Ddc are found in different scaffolds, thus the situation is completely different to the Tenebrio molitor. The authors should show the evidence why Ddc was decreased in TH knock out crickets or add the appropriate discussion.

Minor points

Line 5

"knoc kout" should be "knock out" in the short title .

Materials and Methods

Some values are described with hyphen and others are without. Authors should unify their description without hyphens.

Line 202

Authors explained about gene structure of TH, but did not about yellow-y, although the figure 1 contains both TH and yellow-y.

Line 237

Fig. 7 should be Fig. 8.

Line 248

wild-type should be WT.

Line 263-264

Fig. 11A should be Fig. 11A,B and Fig. 11B should be Fig. 11C.

Line 480

larval instar should be nymphal instar.

Line 495-496

yellow-y should be in italic.

Reviewer #3: (No Response)

7. PLOS authors have the option to publish the peer review history of their article (what does this mean?). If published, this will include your full peer review and any attached files.

Reviewer #1: No

Reviewer #2: No

Reviewer #3: No

---

## [Author Response · Author response to Decision Letter 2]

20 Jan 2023

Reviewer #1: Although the authors responded well to the reviewers’comments, several minor changes are still required.

In Table 1, Please describe the differences of the nucleotide shown capital and small letters.

Please match the decimal place that used in the main text and table.

In Figure 1, UTRs should be UTR.

We thank the reviewer for these suggestions. We have revised them all.

Reviewer #2: The authors reworked Figures properly and modified Tables with appropriate values. The authors cited new references and added discussion, but the research focus of this manuscript was still to establish and optimize the genome editing strategy using Cas9 protein, and the authors did not shift to the research focus to pigmentation process. The genome editing strategy using Cas9 protein were already established as I mentioned through the review comments, so the novelty of this manuscript is none if the authors insist to publish the manuscript as methods paper. In addition, I recommended to add discussion about the Ddc gene expression in previous review comment, but was not enough discussed. I strongly recommend all authors must read and check all the text, figures and tables, because there are lots of careless mistakes in this manuscript.

We thank the reviewer for these very meaningful suggestions. 

We have removed the discussion for methods and the conclusions for methods. We have also made a new combination of the experimental results, which are mainly divided into: Identification and analysis of TH gene and yellow-y gene, The effectiveness of TH gene and yellow-y gene Targeting, Phenotypic analysis and Suppression of TH impairs G. bimaculatus pigmentation and Dopamine synthesis.

We reorganized the abstract and clarified the functions of TH gene and yellow-y gene. “Tyrosine hydroxylase (TH) and yellow-y are involved in insect melanin and the catecholamine biosynthesis pathway. Compared with the control group, mutations of TH and yellow-y genes were defective in pigmentation. The result of quantitative real-time PCR analysis revealed that TH and yellow-y were down-regulated in mutants. Most mutations of the TH gene died by the first instar, and the only adult had significant defects in the wings and legs. Therefore, TH gene is very important for the growth and development of G. bimaculatus,” please see lines 28 to 39 on page 2.

In the introduction, we add the process of melanosis. “The function of TH gene and yellow-y gene was emphasized. Insect cuticle is soft and pale when it is initial synthesized. The stratum corneum hardening process is complex (Andersen, 2012). During the sclerotization process, tyrosine hydroxylase (TH) is the initial rate-limiting synthetic enzyme involved in biosynthesis of 3,4-dihydroxyphenylalanine (DOPA) (Nagatsu et al., 1964; Axelrod, 1971). DOPA is a significant melanin precursor of quinones which are essential for exoskeleton pigmentation and hardening of the cuticle (Hultmark, 1993; Christensen et al., 2005). Ddc, a pyridoxal-5-phosphate-dependent enzyme, catalyzes the conversion of DOPA to dopamine which is an important neuro-transmitter (Eveleth et al., 1986; Scholnick et al., 1986). Insect major royal jelly proteins (MRJP) or Yellow proteins contain an approximately 300 amino acid-long MRJP domain and were initially identified in the royal jelly proteins that play a central role in honeybee development (Albert et al., 1999).” Please see lines 74 to 83 on page 4.

In the result, we combine the original part 2 and Part 4. So the order of the graph is adjusted according to the content. We changed the original Figure 2 into the supplementary Figure 1. We re-matched the sequence and target of yellow-y gene and determined that AGG was PAM sequence, so we re-labeled AGG sequence in Figure 2 and Figure 3. We combined Table 2 and Table 3 and unified the format.

“TH participates in the first step of DOPA production from tyrosine (Futahashi and Fujiwara, 2005; Gorman et al., 2007). In Tenebrio molitor and G. bimaculatus, the Ddc gene is located downstream of the TH gene (Mun et al., 2020; Seike et al., 2022). Similar phenotypes were obtained from RNAi silencing TH gene and Ddc gene expression levels in G. bimaculatus. In this study we found that the expression level of Ddc gene was significantly down-regulated in TH mutants of 1st-instar nymphs. The directly interaction of the two genes in G. bimaculatus still need further verification.” In the follow-up experiment, we will detect the expression levels of TH and Ddc genes at different instars. Please see lines 370 to 377 on page 20.

In a recent study, we confirm that the homozygotes of TH mutants are lethal, so we could not obtain any homozygotes. We preliminarily concluded that TH gene not only affects the pigmentation of G. bimaculatus, but is also crucial for the growth and development of G. bimaculatus. In future studies, we will study the function of TH gene and yellow-y gene respectively, and try to construct double knockout mutants of TH gene and yellow-y gene.

There are the reference of the responses,

Andersen S O. Cuticular sclerotization and tanning[M]//Insect molecular biology and biochemistry. Academic Press, 2012: 167-192.

Nagatsu T, Levitt M, Udenfriend S. Tyrosine hydroxylase: the initial step in norepinephrine biosynthesis. Journal of Biological Chemistry, 1964, 239(9): 2910-2917.

Axelrod J. Noradrenaline: fate and control of its biosynthesis. Science, 1971, 173(3997): 598-606.

Hultmark D. Immune reactions in Drosophila and other insects: a model for innate immunity. Trends in Genetics, 1993, 9(5): 178-183.

Christensen B M, Li J, Chen C C, et al. Melanization immune responses in mosquito vectors. Trends in parasitology, 2005, 21(4): 192-199.

Eveleth D D, Gietz R D, Spencer C A, et al. Sequence and structure of the dopa decarboxylase gene of Drosophila: evidence for novel RNA splicing variants. The EMBO journal, 1986, 5(10): 2663-2672.

Scholnick S B, Bray S J, Morgan B A, et al. CNS and hypoderm regulatory elements of the Drosophila melanogaster dopa decarboxylase gene. Science, 1986, 234(4779): 998-1002.

Albert S, Bhattacharya D, Klaudiny J, et al. The family of major royal jelly proteins and its evolution. Journal of Molecular Evolution, 1999, 49(2): 290-297.

Futahashi R, Fujiwara H. Melanin-synthesis enzymes coregulate stage-specific larval cuticular markings in the swallowtail butterfly, Papilio xuthus. Development genes and evolution, 2005, 215(10): 519-529.

Gorman M J, An C, Kanost M R. Characterization of tyrosine hydroxylase from Manduca sexta. Insect biochemistry and molecular biology, 2007, 37(12): 1327-1337.

Mun S, Noh M Y, Kramer K J, et al. Gene functions in adult cuticle pigmentation of the yellow mealworm, Tenebrio molitor. Insect biochemistry and molecular biology, 2020, 117: 103291.

Seike H, Nagata S. Different transcriptional levels of Corazonin, Elevenin, and PDF according to the body color of the two-spotted cricket, Gryllus bimaculatus. Bioscience, Biotechnology, and Biochemistry, 2022, 86: 23-30.

Major comments:

1. The authors modified figure 3 and figure 9. In these figures, the authors showed that GGC sequences just adjacent to 5' side of sgRNA in yellow-y gene are the PAM sequences, but could be incorrect. PAM sequences must be NGG for Cas9 and locate just adjacent to 3' side of sgRNA, thus, I think that AGG sequences shown in figure 9D could be the PAM sequences. In figure 3D, I was confused that double peaks were appeared far from PAM sequences, as I pointed out in previous review comments. If the AGG was correct PAM sequences, double peaks in figure 3D were very close to PAM sequences and that is reasonable. The authors have to confirm the correct PAM sequences and should rewrite results and discussion and rework many figures, based on the position of PAM sequences.

We re-matched the sequence and target of yellow-y gene and determined that AGG was PAM sequence, so we re-labeled AGG sequence in Figure 2 and Figure 3.

2. To answer my previous comment, the authors discuss about the decreased expression of Ddc in TH knock out crickets in line 332-333, citing the genetic loci of Ddc and TH in Tenebrio molitor. However, in Gryllus bimaculatus, TH and Ddc are found in different scaffolds, thus the situation is completely different to the Tenebrio molitor. The authors should show the evidence why Ddc was decreased in TH knock out crickets or add the appropriate discussion.

TH participates in the first step of DOPA production from tyrosine (Futahashi and Fujiwara, 2005; Gorman et al., 2007). In Tenebrio molitor and G. bimaculatus, the Ddc gene is located downstream of the TH gene (Mun et al., 2020; Seike et al., 2022). Similar phenotypes were obtained from RNAi silencing TH gene and Ddc gene expression levels in G. bimaculatus. In this study we found that the expression level of Ddc gene was significantly down-regulated in TH mutants of 1st-instar nymphs. The directly interaction of the two genes in G. bimaculatus still need further verification. In the follow-up experiment, we will detect the expression levels of TH and Ddc genes at different instars. Please see lines 370 to 377 on page 20.

Futahashi R, Fujiwara H. Melanin-synthesis enzymes coregulate stage-specific larval cuticular markings in the swallowtail butterfly, Papilio xuthus. Development genes and evolution, 2005, 215(10): 519-529.

Gorman M J, An C, Kanost M R. Characterization of tyrosine hydroxylase from Manduca sexta. Insect biochemistry and molecular biology, 2007, 37(12): 1327-1337.

Mun S, Noh M Y, Kramer K J, et al. Gene functions in adult cuticle pigmentation of the yellow mealworm, Tenebrio molitor. Insect biochemistry and molecular biology, 2020, 117: 103291.

Seike H, Nagata S. Different transcriptional levels of Corazonin, Elevenin, and PDF according to the body color of the two-spotted cricket, Gryllus bimaculatus. Bioscience, Biotechnology, and Biochemistry, 2022, 86: 23-30.

Minor points

Line 5

"knoc kout" should be "knock out" in the short title .

Materials and Methods

Some values are described with hyphen and others are without. Authors should unify their description without hyphens.

Line 202

Authors explained about gene structure of TH, but did not about yellow-y, although the figure 1 contains both TH and yellow-y.

Line 237

Fig. 7 should be Fig. 8.

Line 248

wild-type should be WT.

Line 263-264

Fig. 11A should be Fig. 11A,B and Fig. 11B should be Fig. 11C.

Line 480

larval instar should be nymphal instar.

Line 495-496

yellow-y should be in italic.

We thank the reviewer for these suggestions. We have revised them all.

---

## [Decision Letter · Decision Letter 3]

28 Feb 2023

PONE-D-22-04580R3CRISPR/Cas9-Mediated Genomic Knock out of Tyrosine Hydroxylase and Yellow genes in Cricket Gryllus bimaculatusPLOS ONE

Dear Dr. He,

Thank you for submitting your manuscript to PLOS ONE. After careful consideration, we feel that it has merit but does not fully meet PLOS ONE’s publication criteria as it currently stands. Therefore, we invite you to submit a revised version of the manuscript that addresses the points raised during the review process.

One of the reviewers pointed out some incorrectness in the revised manuscript. Please check and revise them, if necessary.

In addition, it comes to our notice that some sentences are significantly or even completely identical to those of other publications. Examples are:

Lines 61 - 63

Lines 71 - 79

Lines 107 - 108

Lines 112 - 115

Lines 121 - 122

Lines 124 - 127

Lines 128 - 129

Lines 159 - 161

Lines 181 - 183

Lines 184 - 193

Lines 264 - 268

Lines 277 - 290

Lines 291 - 293

Lines 314 - 315

Lines 321 - 322

Please rephrase them to avoid potential suspicion of plagiarism, which is not accepted. iThenticate or similar tool should be used to confirm that there are no sentences identical to those of other publications.

We look forward to receiving your revised manuscript.

Kind regards,

Hodaka Fujii, M.D., Ph.D.

Academic Editor

PLOS ONE

Journal Requirements:

Reviewers' comments:

Reviewer's Responses to Questions

**Comments to the Author**

1. If the authors have adequately addressed your comments raised in a previous round of review and you feel that this manuscript is now acceptable for publication, you may indicate that here to bypass the “Comments to the Author” section, enter your conflict of interest statement in the “Confidential to Editor” section, and submit your "Accept" recommendation.

Reviewer #1: All comments have been addressed

Reviewer #2: (No Response)

2. Is the manuscript technically sound, and do the data support the conclusions?

Reviewer #1: Yes

Reviewer #2: Yes

3. Has the statistical analysis been performed appropriately and rigorously? 

Reviewer #1: N/A

Reviewer #2: Yes

4. Have the authors made all data underlying the findings in their manuscript fully available?

Reviewer #1: Yes

Reviewer #2: Yes

5. Is the manuscript presented in an intelligible fashion and written in standard English?

Reviewer #1: Yes

Reviewer #2: Yes

6. Review Comments to the Author

Reviewer #1: All the points raised in the previous version of MS have been addressed by the authors. This reviewer satisfied with the revisions.

Reviewer #2: The authors changed the research focus from the establishment of genome editing strategy to the functional analyses of pigmentation gene. The authors responded reviewers' comments, however, there are still incorrect in supplementary figure, and the authors need to further rework the figure.

Major comment

I pointed out the mistakes of the PAM sequences of yellow-y in my previous comment, and the authors rework the figure 3. However, the PAM sequences of yellow-y indicated in Supplementary Fig 2 were still incorrect. The authors should rework them.

Minor comments

line 205-206:The authors should insert space before the "bp".

line 236 "The legs and wings of the adult is" should be "The legs and wings of the adult are".

7. PLOS authors have the option to publish the peer review history of their article (what does this mean?). If published, this will include your full peer review and any attached files.

Reviewer #1: No

Reviewer #2: No

---

## [Author Response · Author response to Decision Letter 3]

6 Mar 2023

Dear Editor,

We would like to thank you first for all the suggestions of our manuscript (Manuscript Number: PONE-D-22-04580) entitled “CRISPR/Cas9-Mediated Genomic Knock out of Tyrosine Hydroxylase and Yellow genes in Cricket Gryllus bimaculatus”. We really appreciate your help and patience. 

We have seriously thought about the suggestions and provided our response to reviewers. 

It comes to our notice that some sentences are significantly or even completely identical to those of other publications. Examples are:

Lines 61 - 63

Lines 71 - 79

Lines 107 - 108

Lines 112 - 115

Lines 121 - 122

Lines 124 - 127

Lines 128 - 129

Lines 159 - 161

Lines 181 - 183

Lines 184 - 193

Lines 264 - 268

Lines 277 - 290

Lines 291 - 293

Lines 314 - 315

Lines 321 - 322

We thank the reviewer for these suggestions. In the writing process of this article, we mainly refer to the work of students in our research group (Yang, Y., Wang, Y. H., Chen, X. E., Tian, D., Xu, X., Li, K., ... & He, L. (2018). CRISPR/Cas9‐mediated Tyrosine hydroxylase knockout resulting in larval lethality in Agrotis ipsilon. Insect science, 25(6), 1017-1024.). The experimental method is similar to the related research topic in this article. Therefore, some sentences are similar. We have revised them all.

Major comment

I pointed out the mistakes of the PAM sequences of yellow-y in my previous comment, and the authors rework the figure 3. However, the PAM sequences of yellow-y indicated in Supplementary Fig 2 were still incorrect. The authors should rework them.

Minor comments

line 205-206:The authors should insert space before the "bp".

line 236 "The legs and wings of the adult is" should be "The legs and wings of the adult are".

As for the issue in Supplementary Fig 2 raised by reviewer 2, we have revised it, please see Supplementary Fig 2.

We thank the reviewer for these suggestions. We have revised them all.

---

## [Decision Letter · Decision Letter 4]

27 Mar 2023

CRISPR/Cas9-Mediated Genomic Knock out of Tyrosine Hydroxylase and Yellow genes in Cricket Gryllus bimaculatus

PONE-D-22-04580R4

Dear Dr. He,

We’re pleased to inform you that your manuscript has been judged scientifically suitable for publication and will be formally accepted for publication once it meets all outstanding technical requirements.

Kind regards,

Hodaka Fujii, M.D., Ph.D.

Academic Editor

PLOS ONE

Additional Editor Comments (optional):

Reviewers' comments:

Reviewer's Responses to Questions

**Comments to the Author**

1. If the authors have adequately addressed your comments raised in a previous round of review and you feel that this manuscript is now acceptable for publication, you may indicate that here to bypass the “Comments to the Author” section, enter your conflict of interest statement in the “Confidential to Editor” section, and submit your "Accept" recommendation.

Reviewer #1: All comments have been addressed

Reviewer #2: (No Response)

2. Is the manuscript technically sound, and do the data support the conclusions?

Reviewer #1: Yes

Reviewer #2: Yes

3. Has the statistical analysis been performed appropriately and rigorously? 

Reviewer #1: N/A

Reviewer #2: Yes

4. Have the authors made all data underlying the findings in their manuscript fully available?

Reviewer #1: Yes

Reviewer #2: Yes

5. Is the manuscript presented in an intelligible fashion and written in standard English?

Reviewer #1: Yes

Reviewer #2: Yes

6. Review Comments to the Author

Reviewer #1: All the points raised in the previous version of MS have been addressed by the authors.

Reviewer #2: The authors reworked Supplementary Figure 2 and edited some sentences according to the reviewers' comments. The authors rewrote some sentences according to the editors' comments, but the authors need to edit out some minor editing errors.

Minor comments

In these sentences, insert a space before the brackets.

line 75: synthesized(Andersen, 2012).

line 265 rate(Bassett et al., 2013; Bi et al., 2016).

line 287 development(Gorman and Arakane, 2010).

These grammatical careless errors can be prevented by having all co-authors read the manuscript. Authors must make every effort to reduce grammatical errors before submission.

7. PLOS authors have the option to publish the peer review history of their article (what does this mean?). If published, this will include your full peer review and any attached files.

Reviewer #1: No

Reviewer #2: No

---

## [Editor Report · Acceptance letter]

31 Mar 2023

PONE-D-22-04580R4 

CRISPR/Cas9-Mediated Genomic Knock out of Tyrosine Hydroxylase and Yellow Genes in Cricket *Gryllus bimaculatus*

Dear Dr. He:

I'm pleased to inform you that your manuscript has been deemed suitable for publication in PLOS ONE. Congratulations! Your manuscript is now with our production department. 

Kind regards, 

on behalf of

Dr. Hodaka Fujii 

Academic Editor

PLOS ONE